# Learning Gaussian Processes by Minimizing PAC-Bayesian Generalization Bounds

**David Reeb**     **Andreas Doerr**     **Sebastian Gerwinn**     **Barbara Rakitsch**
Bosch Center for Artificial Intelligence*
Robert-Bosch-Campus 1
71272 Renningen, Germany
{david.reeb,andreas.doerr3,sebastian.gerwinn,barbara.rakitsch}@de.bosch.com

## Abstract

Gaussian Processes (GPs) are a generic modelling tool for supervised learning. While they have been successfully applied on large datasets, their use in safety-critical applications is hindered by the lack of good performance guarantees. To this end, we propose a method to learn GPs and their sparse approximations by directly optimizing a PAC-Bayesian bound on their generalization performance, instead of maximizing the marginal likelihood. Besides its theoretical appeal, we find in our evaluation that our learning method is robust and yields significantly better generalization guarantees than other common GP approaches on several regression benchmark datasets.

## 1 Introduction

Gaussian Processes (GPs) are a powerful modelling method due to their non-parametric nature [1]. Although GPs are probabilistic models and hence come equipped with an intrinsic measure of uncertainty, this uncertainty does not allow conclusions about their performance on previously unseen test data. For instance, one often observes overfitting if a large number of hyperparameters is adjusted using marginal likelihood optimization [2]. While a fully Bayesian approach, i.e. marginalizing out the hyperparameters, reduces this risk, it incurs a prohibitive runtime since the predictive distribution is no longer analytically tractable. Also, it does not entail out-of-the-box safety guarantees.

In this work, we propose a novel training objective for GP models, which enables us to give *rigorous* and quantitatively good performance guarantees on future predictions. Such rigorous guarantees are developed within Statistical Learning Theory (e.g. [3]). But as the classical uniform learning bounds are meaningless for expressive models like deep neural nets [4] (as e.g. the VC dimension exceeds the training size) and GPs or non-parametric methods in general, such guarantees cannot be employed for learning those models. Instead, common optimization schemes are (regularized) empirical risk minimization (ERM) [4, 3], maximum likelihood (MLE) [1], or variational inference (VI) [5, 6].

On the other hand, better non-uniform learning guarantees have been developed within the PAC-Bayesian framework [7, 8, 9] (Sect. 2). They are specially adapted to probabilistic methods like GPs and can yield tight generalization bounds, as observed for GP classification [10], probabilistic SVMs [11, 12], linear classifiers [13], or stochastic NNs [14]. Most previous works used PAC-Bayesian bounds merely for the final evaluation of the generalization performance, whereas learning by optimizing a PAC-Bayesian bound has been barely explored [13, 14]. This work, for the first time, explores the use of PAC-Bayesian bounds *(a)* for GP training and *(b)* in the regression setting.

Specifically, we propose to learn full and sparse GP predictors $Q$ directly by minimizing a PAC-Bayesian upper bound $B(Q)$ from Eq. (5) on the true future risk $R(Q)$ of the predictor, as a

principled method to ensure good generalization (Sect. 3). Our general approach comes naturally for GPs because the KL divergence $KL(Q\|P)$ in the PAC-Bayes theorem can be evaluated analytically for GPs $P, Q$ sharing the same hyperparameters. As this applies to popular sparse GP variants such as DTC [16], FITC [15], and VFE [6], they all become amenable to our method of PAC-Bayes learning, combining computational benefits of sparse GPs with theoretical guarantees. We carefully account for the different types of parameters (hyperparameters, inducing inputs, observation noise, free-form parameters), as only some of them contribute to the "penalty term" in the PAC-Bayes bound. Further, we base GP learning directly on the inverse binary KL divergence [10], and not on looser bounds used previously, such as from Pinsker's inequality (e.g., [14]).

We demonstrate our GP learning method on regression tasks, whereas PAC-Bayes bounds have so far mostly been used in a classification setting. A PAC-Bayesian bound for regression with potentially unbounded loss function was developed in [17], it requires a sub-Gaussian assumption w.r.t. the (unknown) data distribution, see also [18]. To remain distribution-free as in the usual PAC setting, we employ and investigate a generic *bounded* loss function for regression.

We evaluate our learning method on several datasets and compare its performance to state-of-the-art GP methods [1, 15, 6] in Sect. 4. Our learning objective exhibits robust optimization behaviour with the same scaling to large datasets as the other GP methods. We find that our method yields significantly better risk bounds, often by a factor of more than two, and that only for our approach the guarantee improves with the number of inducing points.

## 2 General PAC-Bayesian Framework

### 2.1 Risk functions

We consider the standard supervised learning setting [3] where a set $S$ of $N$ training examples $(x_i, y_i) \in X \times Y$ $(i = 1, \ldots, N)$ is used to learn in a hypothesis space $\mathcal{H} \subseteq Y^X$, a subset of the space of functions $X \to Y$. We allow learning algorithms that output a distribution $Q$ over hypotheses $h \in \mathcal{H}$, rather than a single hypothesis $h$, which is the case for GPs we consider later on.

To quantify how well a hypothesis $h$ performs, we assume a bounded loss function $\ell : Y \times Y \to [0, 1]$ to be given, w.l.o.g. scaled to the interval $[0, 1]$. $\ell(y_*, \widehat{y})$ measures how well the prediction $\widehat{y} = h(x_*)$ approximates the actual output $y_*$ at an input $x_*$. The *empirical risk* $R_S(h)$ of a hypothesis is then defined as the average training loss $R_S(h) := \frac{1}{N} \sum_{i=1}^{N} \ell(y_i, h(x_i))$. As in the usual PAC framework, we assume an (unknown) underlying distribution $\mu = \mu(x, y)$ on the set $X \times Y$ of examples, and define the *(true) risk* as $R(h) := \int d\mu(x, y)\ell(y, h(x))$. We will later assume that the training set $S$ consists of $N$ independent draws from $\mu$ and study how close $R_S$ is to its mean $R$ [3]. To quantify the performance of stochastic learning algorithms, that output a distribution $Q$ over hypotheses, we define the empirical and true risks by a slight abuse of notation as [7]:

$$R_S(Q) := \mathbb{E}_{h \sim Q}\big[R_S(h)\big] = \frac{1}{N} \sum_{i=1}^{N} \mathbb{E}_{h \sim Q}\big[\ell\big(y_i, h(x_i)\big)\big], \tag{1}$$

$$R(Q) := \mathbb{E}_{h \sim Q}\big[R(h)\big] = \mathbb{E}_{(x_*, y_*) \sim \mu} \mathbb{E}_{h \sim Q}\big[\ell\big(y_*, h(x_*)\big)\big]. \tag{2}$$

These are the average losses, also termed *Gibbs risks*, on the training and true distributions, respectively, where the hypothesis $h$ is sampled according to $Q$ before prediction.

In the following, we focus on the regression case, where $Y \subseteq \mathbb{R}$ is the set of reals. An exemplary loss function in this case is $\ell(y_*, \widehat{y}) := \mathbb{1}_{\widehat{y} \notin [r_-(y_*), r_+(y_*)]}$, where the functions $r_\pm$ specify an interval outside of which a prediction $\widehat{y}$ is deemed insufficient; similar to $\varepsilon$-support vector regression [19], we use $r_\pm(y_*) := y_* \pm \varepsilon$, with a desired accuracy goal $\varepsilon > 0$ specified before learning (see Sect. 4). In any case, the expectations over $h \sim Q$ in (1)–(2) reduce to one-dimensional integrals as $h(x_*)$ is a real-valued random variable at each $x_*$. See App. C, where we also explore other loss functions.

Instead of the stochastic predictor $h(x_*)$ with $h \sim Q$, one is often interested in the deterministic *Bayes predictor* $\mathbb{E}_{h \sim Q}[h(x_*)]$ [10]; for GP regression, this simply equals the predictive mean $\widehat{m}(x_*)$ at $x_*$. The corresponding *Bayes risk* is defined by $R_{\text{Bay}}(Q) := \mathbb{E}_{(x_*, y_*) \sim \mu}[\ell(y_*, \mathbb{E}_{h \sim Q}[h(x_*)])]$. While PAC-Bayesian theorems do not directly give a bound on $R_{\text{Bay}}(Q)$ but only on $R(Q)$, it is easy to see that $R_{\text{Bay}}(Q) \leq 2R(Q)$ if $\ell(y_*, \widehat{y})$ is quasi-convex in $\widehat{y}$, as in the examples above, and the

distribution of $\widehat{y} = h(x_*)$ is symmetric around its mean (e.g., Gaussian) [10]. An upper bound $B(Q)$ on $R(Q)$ below $1/2$ thus implies a nontrivial bound on $R_{\mathrm{Bay}}(Q) \leq 2B(Q) < 1$.

## 2.2 PAC-Bayesian generalization bounds

In this paper we aim to learn a GP $Q$ by minimizing suitable risk bounds. Due to the probabilistic nature of GPs, we employ generalization bounds for stochastic predictors, which were previously observed to yield stronger guarantees than those for deterministic predictors [10, 11, 14]. The most important results in this direction are the so-called "PAC-Bayesian bounds", originating from [7, 8] and developed in various directions [10, 20, 9, 13, 21, 17].

The PAC-Bayesian theorem (Theorem 1) gives a probabilistic upper bound (generalization guarantee) on the true risk $R(Q)$ of a stochastic predictor $Q$ in terms of its empirical risk $R_S(Q)$ on a training set $S$. It requires to fix a distribution $P$ on the hypothesis space $\mathcal{H}$ *before* seeing the training set $S$, and applies to the true risk $R(Q)$ of *any* distribution $Q$ on $\mathcal{H}^1$. The bound contains a term that can be interpreted as complexity of the hypothesis distribution $Q$, namely the Kullback-Leibler (KL) divergence $KL(Q\|P) := \int dh\, Q(h) \ln \frac{Q(h)}{P(h)}$, which takes values in $[0, +\infty]$. The bound also contains the binary KL-divergence $kl(q\|p) := q \ln \frac{q}{p} + (1-q) \ln \frac{1-q}{1-p}$, defined for $q, p \in [0, 1]$, or more precisely its (upper) inverse $kl^{-1}$ w.r.t. the second argument (for $q \in [0, 1]$, $\varepsilon \in [0, \infty]$):

$$kl^{-1}(q, \varepsilon) := \max\{p \in [0, 1] \,:\, kl(q \,\|\, p) \leq \varepsilon\}, \tag{3}$$

which equals the unique $p \in [q, 1]$ satisfying $kl(q\|p) = \varepsilon$. While $kl^{-1}$ has no closed-form expression, we refer to App. A for an illustration and more details, including its derivatives for optimization.

**Theorem 1** (PAC-Bayesian theorem [7, 10, 20]). *For any $[0, 1]$-valued loss function $\ell$, for any distribution $\mu$, for any $N \in \mathbb{N}$, for any distribution $P$ on a hypothesis set $\mathcal{H}$, and for any $\delta \in (0, 1]$, the following holds with probability at least $1 - \delta$ over the training set $S \sim \mu^N$:*

$$\forall Q: \quad R(Q) \,\leq\, kl^{-1}\left(R_S(Q), \frac{KL(Q \,\|\, P) + \ln \frac{2\sqrt{N}}{\delta}}{N}\right). \tag{4}$$

The RHS of (4) can be upper bounded by $R_S(Q) + \sqrt{\left(KL(Q \,\|\, P) + \ln \frac{2\sqrt{N}}{\delta}\right)/(2N)}$, which gives a useful intuition about the involved terms, but can exceed 1 and thereby yield a trivial statement. Note that the full PAC-Bayes theorem [20] gives a simultaneous lower bound on $R(Q)$, which is however not relevant here as we are going to *minimize* the upper risk bound. Further refinements of the bound are possible (e.g., [20]), but as they improve over Theorem 1 only in small regimes [9, 13, 21], often despite adjustable parameters, we will stick with the parameter-free bound (4).

We want to consider a family of prior distributions $P^\theta$ parametrized by $\theta \in \Theta$, e.g. in GP hyperparameter training [1]. If this family is countable, one can generalize the above analysis by fixing some probability distribtion $p_\theta$ on $\Theta$ and defining the mixture prior $P := \sum_\theta p_\theta P^\theta$; when $\Theta$ is a finite set, the uniform distribution $p_\theta = 1/|\Theta|$ is a canonical choice. Using the fact that $KL(Q \,\|\, P) \leq KL(Q \,\|\, P^\theta) + \ln \frac{1}{p_\theta}$ holds for each $\theta \in \Theta$ (App. B), Theorem 1 yields that, with probability at least $1 - \delta$ over $S \sim \mu^N$,

$$\forall \theta \in \Theta\, \forall Q: \quad R(Q) \leq kl^{-1}\left(R_S(Q), \frac{KL(Q \,\|\, P^\theta) + \ln \frac{1}{p_\theta} + \ln \frac{2\sqrt{N}}{\delta}}{N}\right) \;=:\; B(Q). \tag{5}$$

The bound (5) holds simultaneously for all $P^\theta$ and all $Q^2$. One can thus optimize over both $\theta$ and $Q$ to obtain the best generalization guarantee, with confidence at least $1 - \delta$. We use $B(Q)$ for our training method below, but we will also compare to training with the suboptimal upper bound $B_{\mathrm{Pin}}(Q) := R_S(Q) + \sqrt{\left(KL(Q\|P^\theta) + \ln \frac{1}{p_\theta} + \ln \frac{2\sqrt{N}}{\delta}\right)/(2N)} \geq B(Q)$ as was done previously [14]. The PAC-Bayesian bound depends only weakly on the confidence parameter $\delta$, which enters

logarithmically and is suppressed by the sample size $N$. When the hyperparameter set $\Theta$ is not too large (i.e. $\ln \frac{1}{p_\theta}$ is small compared to $N$), the main contribution to the *penalty term* in the second argument of $kl^{-1}$ comes from $\frac{1}{N}KL(Q\|P^\theta)$, which must be $\ll 1$ for a good generalization statement (see Sect. 4).

## 3 PAC-Bayesian learning of GPs

### 3.1 Learning full GPs

GP modelling is usually presented as a Bayesian method [1], in which the prior $P(f) = \mathcal{GP}(f|m(x), K(x,x'))$ is specified by a positive definite kernel $K : X \times X \to \mathbb{R}$ and a mean function $m : X \to \mathbb{R}$ on the input set $X$. In ordinary GP regression, the learned distribution $Q$ is then chosen as the Bayesian posterior coming from the assumption that the training outputs $y_N := (y_i)_{i=1}^N \in \mathbb{R}^N$ are noisy versions of $f_N = (f(x_1), \ldots, f(x_N))$ with i.i.d. Gaussian likelihood $y_N|f_N \sim \mathcal{N}(y_N|f_N, \sigma_n^2 \mathbb{1})$. Under this assumption, $Q$ is again a GP [1]:

$$
\begin{aligned}
Q(f) = \ &\mathcal{GP}\big(f \mid m(x) + k_N(x)(K_{NN} + \sigma_n^2 \mathbb{1})^{-1}(y_N - m_N), \\
&K(x,x') - k_N(x)(K_{NN} + \sigma_n^2 \mathbb{1})^{-1}k_N(x')^T\big),
\end{aligned}
\tag{6}
$$

with $K_{NN} = (K(x_i, x_j))_{i,j=1}^N, k_N(x) = (K(x, x_1), \ldots, K(x, x_N)), m_N = (m(x_1), \ldots, m(x_N))$. Eq. (6) is employed to make (stochastic) predictions for $f(x_*)$ on new inputs $x_* \in X$. In our approach below, we do not require any Bayesian rationale behind $Q$ but merely use its form, parametrized by $\sigma_n^2$, as an optimization ansatz within the PAC-Bayesian theorem.

Importantly, for any full GP prior $P$ and its corresponding posterior $Q$ from (6), the KL-divergence $KL(Q\|P)$ in Theorem 1 and Eq. (5) can be evaluated on *finite* ($N$-)dimensional matrices. This allows us to evaluate the PAC-Bayesian bound and in turn to learn GPs by optimizing it. More precisely, one can easily verify that $P$ and $Q$ have the *same* conditional distribution $P(f|f_N) = Q(f|f_N)^3$, so that

$$
KL(Q \| P) = KL(Q(f_N)Q(f \mid f_N) \| P(f_N)P(f \mid f_N)) = KL(Q(f_N) \| P(f_N)) \tag{7}
$$

$$
= \frac{1}{2}\ln\det\big[K_{NN} + \sigma_n^2\mathbb{1}\big] - \frac{N}{2}\ln\sigma_n^2 - \frac{1}{2}\mathrm{tr}\big[K_{NN}(K_{NN} + \sigma_n^2\mathbb{1})^{-1}\big] \tag{8}
$$

$$
+ \frac{1}{2}(y_N - m_N)^T(K_{NN} + \sigma_n^2\mathbb{1})^{-1}K_{NN}(K_{NN} + \sigma_n^2\mathbb{1})^{-1}(y_N - m_N),
$$

where in the last step we used the well-known formula [22] for the KL divergence between normal distributions $P(f_N) = \mathcal{N}(f_N \mid m_N, K_{NN})$ and $Q(f_N) = \mathcal{N}\big(f_N \mid m_N + K_{NN}(K_{NN} + \sigma_n^2\mathbb{1})^{-1}(y_N - m_N), K_{NN} - K_{NN}(K_{NN} + \sigma_n^2\mathbb{1})^{-1}K_{NN}\big)$, and simplified a bit (see also App. D).

To learn a full GP means to select "good" values for the *hyperparameters* $\theta$, which parametrize a family of GP priors $P^\theta = \mathcal{GP}(f|m^\theta(x), K^\theta(x,x'))$, and for the noise level $\sigma_n$ [1]. Those values are afterwards used to make predictions with the corresponding posterior $Q^{\theta,\sigma_n}$ from (6). In our experiments (Sect. 4) we will use the squared exponential (SE) kernel on $X = \mathbb{R}^d$, $K^\theta(x,x') = \sigma_s^2\exp[-\frac{1}{2}\sum_{i=1}^d \frac{(x_i - x_i')^2}{l_i^2}]$, where $\sigma_s^2$ is the signal variance, $l_i$ are the lengthscales, and we set the mean function to zero. The hyperparameters are $\theta \equiv (l_1^2, \ldots, l_d^2, \sigma_s^2)$ (SE-ARD kernel [1]), or $\theta \equiv (l^2, \sigma_s^2)$ if we take all lengthscales $l_1 = \ldots = l_d \equiv l$ to be equal (non-ARD).

The basic idea of our method, which we call "PAC-GP" is now to learn the parameters[4] $\theta$ and $\sigma_n$ by minimizing the upper bound $B(Q^{\theta,\sigma_n})$ from Eq. (5), therefore selecting the GP predictor $Q^{\theta,\sigma_n}$ with the best generalization performance guarantee within the scope of the PAC-Bayesian bound. Note

that all involved terms $R_S(Q^{\theta,\sigma_n})$ (App. C) and $KL(Q^{\theta,\sigma_n}\|P^\theta)$ from (8) as well as their derivatives (App. A) can be computed effectively, so we can use gradient-based optimization.

The only remaining issue is that the learned prior hyperparameters $\theta$ have to come from a discrete set $\Theta$ that must be specified before seeing the training set $S$ (Sect. 2.2). To achieve this, we first minimize the RHS of Eq. (5) over $\theta$ and $\sigma_n^2$ in a gradient-based manner, and thereafter discretize each of the components of $\ln\theta$ to the closest point in the equispaced $(G+1)$-element set $\{-L, -L + \frac{2L}{G}, \ldots, +L\}$; thus, when $T$ denotes the number of components of $\theta$, the penalty term to be used in the optimization objective (5) is $\ln\frac{1}{p_\theta} = \ln|\Theta| = T\ln(G+1)$. The SE-ARD kernel has $T = d+1$, while the standard SE kernel has $T = 2$ parameters. In our experiments we round each component of $\ln\theta$ to two decimal digits in the range $[-6, +6]$, i.e. $L = 6$, $G = 1200$. We found that this discretization has virtually no effect on the predictions of $Q^{\theta,\sigma_n}$, and that coarser rounding (i.e. smaller $|\Theta|$) does not significantly improve the bound (5) (via its smaller penalty term $\ln|\Theta|$) nor the optimization (via its higher sensitivity to $Q$); see App. F.

## 3.2 Learning sparse GPs

Despite the fact that, with confidence $1-\delta$, the bound in (5) holds for *any* $P_\theta$ from the prior GP family and for *any* distribution $Q$, we optimized in Sect. 3.1 the upper bound merely over the parameters $\theta, \sigma_n$ after substituting $P^\theta$ and the corresponding $Q^{\theta,\sigma_n}$ from (6). We are limited by the need to compute $KL(Q\|P)$ effectively, for which we relied on the property $Q(f \mid f_N) = P(f \mid f_N)$ and the Gaussianity of $P(f_N)$ and $Q(f_N)$, cf. (7). Building on this two requirements, we now construct more general pairs $P, Q$ of GPs with effectively computable $KL(Q\|P)$, so that our learning method becomes more widely applicable, including sparse GP methods.

Instead of the points $x_1, \ldots, x_N$ associated with the training set $S$ as in Sect. 3.1, one may choose from the input space any number $M$ of points $Z = \{z_1, \ldots, z_M\} \subseteq X$, often called *inducing inputs*, and any Gaussian distribution $Q(f_M) = \mathcal{N}(f_M \mid a_M, B_{MM})$ on function values $f_M := (f(z_1), \ldots, f(z_M))$, with any $a_M \in \mathbb{R}^M$ and positive semidefinite matrix $B_{MM} \in \mathbb{R}^{M \times M}$. The distribution $Q$ on $f_M$ can be extended to all function values at all inputs $X$ using the conditional $Q(f \mid f_M) = P(f \mid f_M)$ from the prior $P$ (see Sect. 3.1). This yields the following predictive GP:

$$Q(f) = \mathcal{GP}\big(f \mid m(x) + k_M(x)K_{MM}^{-1}(a_M - m_M), \tag{9}$$
$$K(x, x') - k_M(x)K_{MM}^{-1}[K_{MM} - B_{MM}]K_{MM}^{-1}k_M(x')^T\big),$$

where $K_{MM} := (K(z_i, z_j))_{i,j=1}^M$, $k_M(x) := (K(x, z_1), \ldots, K(x, z_M))$, and $m_M := (m(z_1), \ldots, m(z_M))$. This form of $Q$ includes several approximate posteriors from Bayesian inference that have been used in the literature [1, 10, 6, 26], even for noise models other than the Gaussian one used to motivate the $Q$ from Sect. 3.1. Analogous reasoning as in (7) now gives [10, 6, 23]:

$$KL(Q \| P) = KL(Q(f_M) \| P(f_M)) = -\frac{1}{2}\ln\det\big[B_{MM}K_{MM}^{-1}\big] + \frac{1}{2}\mathrm{tr}\big[B_{MM}K_{MM}^{-1}\big] - \frac{M}{2}$$
$$+ \frac{1}{2}(a_M - m_M)^T K_{MM}^{-1}(a_M - m_M). \tag{10}$$

One can thus effectively optimize in (5) the prior $P^\theta$ and the posterior distribution $Q^{\theta,\{z_i\},a_M,B_{MM}}$ by varying the number $M$ and locations $z_1, \ldots, z_M$ of inducing inputs and the parameters $a_M$ and $B_{MM}$, along with the hyperparameters $\theta$. Optimization can in this framework be organized such that it consumes time $O(NM^2 + M^3)$ per gradient step and memory $O(NM + M^2)$ as opposed to $O(N^3)$ and $O(N^2)$ for the full GP of Sect. 3.1. This is a big saving when $M \ll N$ and justifies the name "sparse GP" [1, 24].

Some popular sparse-GP methods [24] are special cases of the above form, by prescribing certain $a_M$ and $B_{MM}$ depending on the training set $S$, so that only the inducing inputs $z_1, \ldots, z_M$ and a few parameters such as $\sigma_n^2$ are left free:

$$a_M = K_{MM}Q_{MM}^{-1}K_{MN}(\alpha\Lambda + \sigma_n^2\mathbb{1})^{-1}y_N, \quad B_{MM} = K_{MM}Q_{MM}^{-1}K_{MM}, \tag{11}$$

where $Q_{MM} = K_{MM} + K_{MN}(\alpha\Lambda + \sigma_n^2\mathbb{1})^{-1}K_{NM}$ with $K_{MN} := (K(z_i, x_j))_{i,j=1}^{M,N}$, $K_{NM} = K_{MN}^T$, and $\Lambda = \mathrm{diag}(\lambda_1, \ldots, \lambda_N)$ is a diagonal $N \times N$-matrix with entries $\lambda_i = K(x_i, x_i) - k_M(x_i)K_{MM}^{-1}k_M(x_i)^T$. Setting $\alpha = 1$ corresponds to the FITC approximation [15], whereas $\alpha = 0$

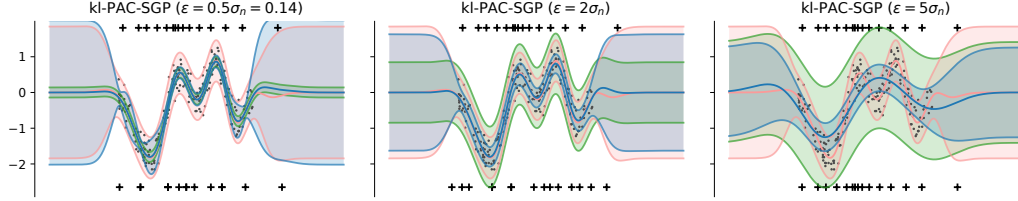

Figure 1: **Predictive distributions.** The predictive distributions (mean $\pm 2\sigma$ as shaded area) of our kl-PAC-SGP (blue) are shown for various choices of $\varepsilon$ together with the full-GP's prediction (red). (Note that by Eqs. (9,11), kl-PAC-SGP's predictive variance does *not* include additive $\sigma_n^2$, whereas full-GP's does [1].) The shaded green area visualizes an $\varepsilon$-band, centered around the kl-PAC-SGP's predictive mean; datapoints (black dots) inside this band do not contribute to the risk $R_S(Q)$. Crosses above/below the plots indicate the inducing point positions ($M = 15$) before/after training.

is the VFE and DTC method [6, 16] (see App. D for their training objectives); one can also linearly interpolate between both choices with $\alpha \geq 0$ [25]. Another form of sparse GPs where the latent function values $f_M$ are fixed and not marginalized over, corresponds to $B_{MM} = 0$, which however gives diverging $KL(Q \parallel P) = \infty$ via (10) and therefore trivial bounds in (4)–(5).

Our learning method for sparse GPs ("PAC-SGP") follows now similar steps as in Sect. 3.1: One has to include a penalty $\ln \frac{1}{p_\theta} = \ln |\Theta|$ for the prior hyperparameters $\theta$, which are to be discretized into the set $\Theta$ after the optimization of (5). Note, $\theta$ contains the prior hyperparameters only and not the inducing points $z_1, \ldots, z_M$ nor $a_M, B_{MM}, \sigma_n$, or $\alpha$ from (11); all these quantities can be optimized over simultaneously with $\theta$, but do not need to be discretized. The number $M$ of inducing inputs can also be varied, which determines the required computational effort, and all optimizations can be both discrete [16] or continuous [15, 6]. When optimizing over positive $B_{MM}$, the parametrization $B_{MM} = LL^T$ with a lower triangular matrix $L \in \mathbb{R}^{M \times M}$ can be used [26]. For the experiments below we always employ the FITC parametrization (fixed $\alpha = 1$) in our proposed PAC-SGP method, i.e. our optimization parameters are $\sigma_n^2$ and $\{z_i\}$ besides the length scale hyperparameters $\theta$.

# 4 Experiments[5]

We now illustrate our learning method and compare it with other GP methods on various regression tasks. In contrast to prior work [14], we found the gradient-based training with the objective (5) to be robust enough, such that no pretraining with conventional objectives (such as from App. D) is necessary. We set $\delta = 0.01$ throughout [10, 14], cf. Sect. 2.2, and use (unless specified otherwise) the generic bounded loss function $\ell(y, \widehat{y}) = \mathbb{1}_{\widehat{y} \notin [y-\varepsilon, y+\varepsilon]}$ for regression, with accuracy goal $\varepsilon > 0$ as specified below.

We evaluate the following methods: *(a) PAC-GP:* Our proposed method (cf. Sect. 3.1) with the training objective $B(Q)$ (5) (kl-PAC-GP) and for comparison with the looser training objective $B_{\mathrm{Pin}}$ (sqrt-PAC-GP) (see below (5), similar to e.g. [14]); *(b) PAC-SGP:* Our sparse GP method (Sect. 3.2), again with objectives $B(Q)$ (kl-PAC-SGP) and $B_{\mathrm{Pin}}(Q)$ (sqrt-PAC-SGP), respectively; *(c) full-GP:* The ordinary full GP for regression [1]; *(d) VFE:* Titsias' sparse GP [6]; *(e) FITC:* Snelson-Ghahramani's sparse GP [15]. Note that full-GP, VFE, and FITC as well as sqrt-PAC-GP and sqrt-PAC-SGP are *trained* on other objectives (see App. D), and we will *evaluate* the upper bound (5) on their generalization performance by evaluating $KL(Q \parallel P)$ via (8) or (10). To obtain finite generalization bounds, we discretize $\theta$ for all methods at the end of training as in Sect. 3.1 and use the appropriate $\ln \frac{1}{p_\theta} = \ln |\Theta|$ in (5).

**(a) Predictive distribution.** To get a first intuition, we illustrate in Fig. 1 the effect of varying $\varepsilon$ in the loss function on the predictive distribution of our sparse PAC-SGP. The accuracy goal $\varepsilon$ defines a band around the predictive mean within which data-points do not contribute to the empirical risk $R_S(Q)$. We thus chose the accuracy goal $\varepsilon$ relative to the observation noise $\sigma_n$ obtained from an

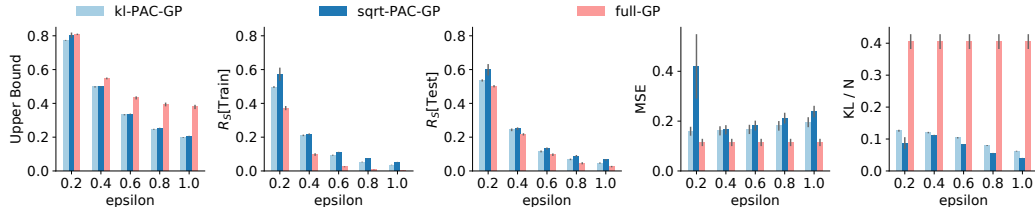

Figure 2: **Dependence on the accuracy goal** $\varepsilon$. For each $\varepsilon$, the plots from left to right show (means as bars, standard errors after ten iterations as grey ticks) the upper bound $B(Q)$ from Eq. (5), the Gibbs training risk $R_S(Q)$, the Gibbs test risk as a proxy for the true $R(Q)$, MSE, and $KL(Q \parallel P^\theta)/N$, after learning $Q$ on the dataset *boston housing* by three different methods: our kl-PAC-GP method from Sect. 3.1 with sqrt-PAC-GP and the ordinary full-GP.

ordinary full-GP. Results are presented on the 1D toy dataset[6] from the original FITC [15] and VFE [6] publications (for a comparison to the predictive distributions of FITC and VFE see App. E, which also contains an illustration that our kl-PAC-SGP avoids FITC's known overfitting on pathological datasets.). Here and below, we optimize the hyperparameters in each experiment anew.

We find that for large $\varepsilon$ (right plot) the predictive distribution (blue) becomes smoother: Due to the wider $\varepsilon$-band (green), the PAC-SGP does not need to adapt much to the data for the $\varepsilon$-band to contain many data points. Hence the predictive distribution can remain closer to the prior, which reduces the KL-term in the objective (5). For the same reason, the inducing points need not adapt much compared to their initial positions for large $\varepsilon$. For smaller $\varepsilon$, the PAC-SGP adapts more to the data, whereas for very small $\varepsilon$ (left plot), it is anyhow not possible to place many data points within the narrow $\varepsilon$-band, so the predictive distribution can again be closer to the prior (compare e.g. in the first and second plots the blue curves near the rightmost datapoints) for a smaller KL-term. In particular, the KL-divergence (divided by number of training points) for the three settings in Fig.1 are: 0.097 (left), 0.109 (middle), and 0.031 (right).

**(b) Full-GP experiments – dependence on the accuracy goal $\varepsilon$.** To explore the dependence on the desired accuracy $\varepsilon$ further, we compare in Fig. 2 the ordinary full-GP to our PAC-GPs on the *boston housing dataset*[7]. As pre-processing we normalized all features and the output to mean zero and unit variance, then analysed the impact of the accuracy goal $\varepsilon \in \{0.2, 0.4, 0.6, 0.8, 1.0\}$. We used 80% of the dataset for training and 20% for testing, in ten repetitions of the experiment.

Our PAC-GP yields significantly better generalization guarantees for all accuracy goals $\varepsilon$ compared to full-GP, since we are directly optimizing the bound (5). This effect is stronger for large $\varepsilon$, where the KL-term of PAC-GP can decrease as $Q$ may again remain closer to $P$ while keeping the training loss low. Although better bounds do not necessarily imply better Gibbs test risk, kl-PAC-GP performs only slightly below the ordinary full-GP in this regard. Moreover, our PAC-GPs exhibit less overfitting than the full-GP, for which the training risks are significantly larger than the test risks (see Table 1 in App. G for numerical values). On the other hand, the tighter objective (5) in the kl-PAC-GP allows learning a slightly more complex GP $Q$ in terms of the KL-divergence compared to the sqrt-PAC-GP, which results in better test risks and at the same time better guarantees. This confirms that kl-PAC-GP is always preferable to sqrt-PAC-GP. However, as any prediction within the full $\pm\varepsilon$-band around the ground truth incurs no risk for our PAC-GPs, their mean squared error (MSE) increases with $\varepsilon$.

The fact that our learned PAC-GPs exhibit higher training and test errors (Gibbs risk and esp. MSE) than full-GP can be explained by their *under*fitting in order to hedge against violating Eq. (5) (i.e. Theorem 1). This underfitting is evidenced by PAC-GP's significantly less complex learned posterior $Q$ as measured by $KL(Q\|P^\theta)/N$ (Fig. 2), or similarly (via Eqs. (8,10)), by its larger learned noise variance $\sigma_n^2$ compared to full-GP's (Table 1 in App. G). It is exactly this stronger regularization of PAC-GP in terms of the $KL$ divergence that leads to its better generalization guarantees.

In the following, we will fix $\varepsilon = 0.6$ after pre-processing data as above, to illustrate PAC-GP further. Note however that in a concrete application, $\varepsilon$ should be fixed to a desired accuracy goal using domain

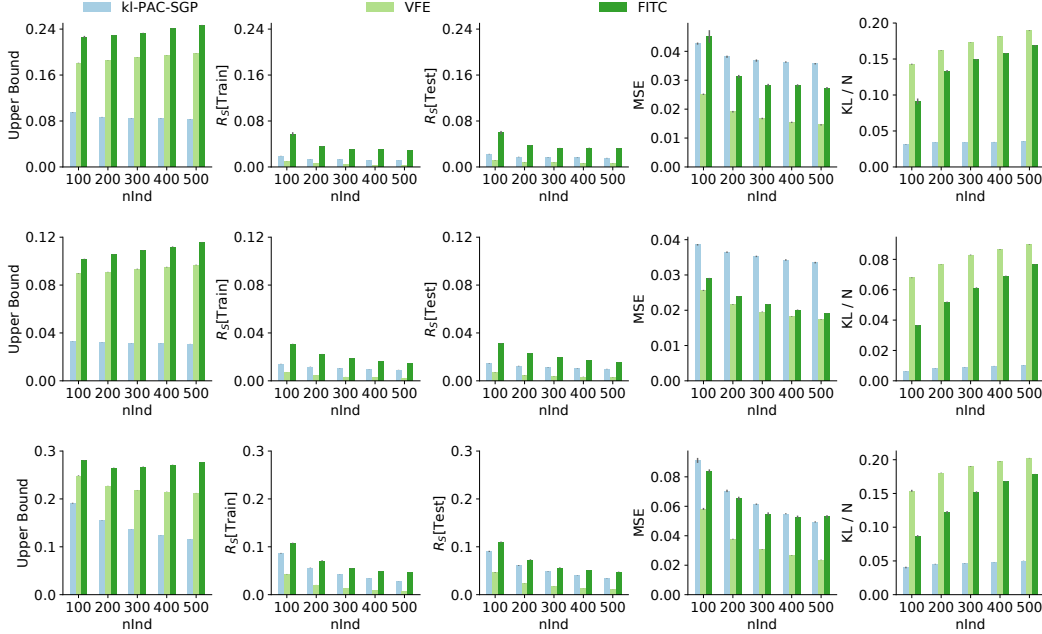

Figure 3: **Dependence on the number of inducing variables.** Shown is the average ($\pm$ standard error over 10 repetitions) upper bound $B$, Gibbs training risk $R_S$, Gibbs test risk, MSE, and $KL(Q\|P^\theta)/N$ as a function of the number $M$ of inducing inputs (from left to right). We compare our sparse kl-PAC-SGP (Sect. 3.2) with the two popular GP approximations VFE and FITC. Each row corresponds to one dataset: *pol* (top), *sarcos* (middle) and *kin40k* (bottom). kl-PAC-SGP has the best guarantee in all settings (left column), due to a lower model complexity (right column), but this comes at the price of slightly larger test errors.

knowledge, but before seeing the training set $S$. Alternatively, one can consider a set of $\varepsilon$-values $\varepsilon_1, \ldots, \varepsilon_E$ chosen in advance, at the cost of a term $\ln E$ in addition to $\log \frac{1}{p_\theta}$ in the objective (5).

**(c) Sparse-GP experiments – dependence on number of inducing inputs $M$.** We now examine our sparse PAC-SGP method (Sect. 3.2) on the three large data sets *pol, sarcos*, and *kin40k*[8], again using 80%–20% train-test splits and ten iterations. The results are shown in Fig. 3. Here, we vary the number of inducing points $M \in \{100, 200, 300, 400, 500\}$. For modelling *pol* and *kin40k*, we use the SE-ARD kernel due to its better performance on these datasets, whereas we model *sarcos* without ARD (cf. Table 2 in App. G for the comparison ARD vs. non-ARD). The corresponding penalty terms for the three plots are $\frac{1}{N} \ln |\Theta| = 0.0160, 0.0003, 0.0020$ and $\frac{1}{N} \ln \frac{2\sqrt{N}}{\delta} = 0.0008, 0.0003, 0.0003$; when compared to $KL(Q\|P)/N$ from Fig. 3, their contribution is largest for the *pol* dataset.

Our kl-PAC-SGP achieves significantly better upper bounds than VFE and FITC, by more than a factor of 3 on *sarcos*, a factor of roughly 2 on *pol*, and a factor between 1.3 and 2 on *kin40k* (Fig. 3, cf. also Table 2 in App. G). Also, the PAC-Bayes upper bound is much tighter for kl-PAC-SGP than for VFE or FITC, i.e. closer to the Gibbs risk, often by factors exceeding 3. Our kl-PAC-SGP behaves also more favorably in terms of generalization guarantee when inducing points are added and more complex models are allowed: our upper bound improves substantially with $M$ (*kin40*) or does at least not degrade (*pol* and *sarcos*), as opposed to VFE and FITC, whose complexities $KL/N$ grow substantially with $M$. Since very low training risks can already be achieved by a moderate number of inducing points for *pol* and *sarcos*, a growing $KL$ with $M$ deteriorates the upper bound. Regarding the upper bound, the increased flexibility from larger $M$ only pays off for the *kin40k* dataset, whereas the MSE improves with increasing $M$ for all models and datasets. As above, kl-PAC-SGP is always slightly preferrable to sqrt-PAC-SGP, not only for the upper bound and Gibbs risks as expected but also for MSE (see Table 2 in App. G).

Similarly to the boston dataset, the higher test errors of kl-PAC-SGP compared to VFE and FITC can be explained by underfitting due to the stronger regularization, again shown by lower $KL$ and significantly larger learned $\sigma_n^2$ (by factors of 4–28 compared to VFE), cf. Table 2 in App. G. In fact, although our implementation of PAC-SGP employs the FITC parametrization, the PAC-(S)GP optimization is not prone to FITC's well-known overfitting tendency [2], due to the regularization via the $KL$-divergence (see App. E, and in particular Supplementary Figure 7).

To investigate whether the higher test MSE of PAC-GP compared to VFE and FITC (and the full-GP above) is a consequence of the 0-1-loss $\ell(y, \widehat{y}) = \mathbb{1}_{\widehat{y} \notin [y-\varepsilon, y+\varepsilon]}$ used so far, we re-ran the PAC-SGP experiments for $M = 500$ inducing inputs with the more distance-sensitive loss function $\ell_{\exp}(y, \widehat{y}) = 1 - \exp[-((y - \widehat{y})/\varepsilon)^2]$ (Eq. (20)), which is MSE-like for small deviations $|y - \widehat{y}| \lesssim \varepsilon$, i.e. $\ell_{\exp}(y, \widehat{y}) \approx (y - \widehat{y})^2/\varepsilon^2$ (Supplementary Figure 5). Our results are tabulated in Table 3 in App. G. The findings are inconclusive and range from an improvement w.r.t. MSE of 25% (*pol*) over little change (*sarcos*) to a decline of 12% (*kin40k*), showing that the effect of the loss function is smaller than might have been expected. Nevertheless, generalization guarantees of PAC-SGP remain much better than the ones of the other methods. While the MSE of our PAC-GPs would improve by choosing smaller $\varepsilon$ (e.g., Fig. 2), this comes at the disadvantage of worse generalization bounds.

We further note that no method shows significant overfitting in Fig. 3, in the sense that the differences between test and training Gibbs risks are all rather small, despite the KL-complexity increasing with $M$ for VFE and FITC. This is unlike for *Boston housing* above, and may be due to the much larger training sets here. When comparing VFE and FITC, we observe that VFE consistently outperforms FITC in terms of both MSE as well as generalization guarantee, where VFE's higher KL-complexity is offset by its much lower Gibbs risk. This fortifies the results in [2]. We lastly note that, since for our PAC-SGP the obtained guarantees $B$ are much smaller than 1/2, we obtain strong guarantees even on the Bayes risk $R_{\text{Bay}} \leq 2B < 1$ (Sect. 2.1).

## 5 Conclusion

In this paper, we proposed and explored the use of PAC-Bayesian bounds as an optimization objective for GP training. Consequently, we were able to achieve significantly better guarantees on the out-of-sample performance compared to state-of-the-art GP methods, such as VFE or FITC, while maintaining computational scalability. We further found that using the tighter generalization bound $B(Q)$ (5) based on the inverse binary kl-divergence leads to an increase in the performance on all metrics compared to a looser bound $B_{\text{Pin}}$ as employed in previous works (e.g. [14]).

Despite the much better generalization guarantees obtained by our method, it often yields worse test error, in particular test MSE, than standard GP regression methods; this largely persists even when using more distance-sensitive loss functions than the 0-1-loss. The underlying reason could be that all loss functions considered in this work were bounded, as necessitated by our desire to provide generalization guarantees irrespective of the true data distribution. While rigorous PAC-Bayesian bounds exist for MSE-like unbounded loss functions under special assumptions on the data distribution [17], it may nevertheless be worthwhile to investigate whether these training objectives lead to better test MSE in examples. A drawback is that those assumptions are usually impossible to verify, thus the generalization guarantees are not comparable. Note that the design of a loss function is dependent on the application domain and there is no ubiquitous choice across all settings. In many safety-critical applications, small deviations are tolerable whereas larger deviations are all equally catastrophic, thus a 0-1-loss as ours and a rigorous bound on it can be more useful than the MSE test error.

While in this work we focussed on regression tasks, the same strategy of optimizing a generalization bound can also be applied to learn GPs for binary and categorical outputs. Note that the true $KL$-term in this setting has so far been merely upper bounded by its regression proxy [10], and it would be interesting to develop better bounds on the classification complexity term. Lastly, it may be worthwhile to use other or more general sparse GPs within our PAC-Bayesian learning method, such as free-form [26] or even more general GPs [29].

### Acknowledgments

We would like to thank Duy Nguyen-Tuong, Martin Schiegg, and Michael Schober for helpful discussions and proofreading.

## Footnotes

*https://www.bosch-ai.com

[1] We follow common usage and call $P$ and $Q$ "prior" and "posterior" distributions in the PAC-Bayesian setting, although their meaning is somewhat different from priors and posteriors in Bayesian probability theory.

[2] The same result can be derived from (4) via a union bound argument (see Appendix B).

[3]In fact, direct computation [1] gives $P(f|f_N) = \mathcal{GP}\big(f|m(x) + k_N(x)K_{NN}^{-1}(f_N - m_N), K(x,x') - k_N(x)K_{NN}^{-1}k_N(x')^T\big) = Q(f|f_N)$. Remarkably, $Q(f|f_N)$ does *not* depend on $y_N$ nor on $\sigma_n$, even though $Q(f)$ from (6) does. Intuitively this is because, for the above likelihood, $f$ is independent of $y_N$ given $f_N$.

[4]Contrary to the usual GP viewpoint [1], $\sigma_n$ is *not* a hyperparameter in our method since the prior $P^\theta$ does not depend on $\sigma_n$. Thus, $\sigma_n$ does also not contribute to the "penalty term" $\ln|\Theta|$. $\sigma_n$ is merely a free parameter in the posterior distribution $Q^{\theta,\sigma_n}$. By (8), $KL(Q^{\theta,\sigma_n}\|P^\theta) \to \infty$ as $\sigma_n \to 0$, so we need this parameter $\sigma_n^2 > 0$ because otherwise $KL = \infty$ and the bound as well as the optimization objective would become trivial. Although the parameter $\sigma_n^2$ is originally motivated by a Gaussian observation noise assumption, the aim here is merely to parameterize the posterior in some way while maintaining computational tractability; cf. also Sect. 3.2.

[5]Python code (building on GPflow [27] and TensorFlow [28]) implementing our method is available at `https://github.com/boschresearch/PAC_GP`.

[6]snelson: dimensions $200 \times 1$, available at `www.gatsby.ucl.ac.uk/~snelson`.

[7]boston: dimensions $506 \times 13$, available at `http://lib.stat.cmu.edu/datasets/boston`

[8]pol: 15,000 $\times$ 26, kin40k: 40,000 $\times$ 8 (both from `https://github.com/trungngv/fgp.git`); sarcos: 48,933 $\times$ 21 (`http://www.gaussianprocess.org/gpml/data`)

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
