[Supplementary Material · gp-pac-bayes-neurips-supplementary.pdf]

Supplementary material for

# Learning Gaussian Processes by Minimizing Generalization Bounds

## A   Inverse binary KL-divergence and its derivatives, Pinsker's inequality

The function $kl^{-1}(q,\varepsilon) \in [q,1]$, defined in Eq. (3), can easily be computed numerically for any $q \in [0,1)$, $\varepsilon \in [0,\infty)$ to any desired accuracy $\Delta > 0$ via the bisection method, since the function $kl(q \parallel p) = q \ln \frac{q}{p} + (1-q) \ln \frac{1-q}{1-p}$ is strictly monotonically increasing in $p \in [q,1]$ from 0 to $\infty$ (for $q = 1$ or $\varepsilon = \infty$, we set $kl^{-1}(q,\varepsilon) := 1$). Note that the monotonicity in $p$ implies further that $kl^{-1}(q,\varepsilon)$ is monotonically increasing in $\varepsilon \in [0,\infty]$. Fig. 4 shows a plot of $kl^{-1}(q,\varepsilon)$ for various values of $\epsilon \geq 0$, and states a few special function values of $kl^{-1}(q,\varepsilon)$. By Pinsker's inequality, it holds that $kl(q \parallel p) \geq 2|p-q|^2$, which implies that $kl^{-1}(q,\varepsilon) \leq q + \sqrt{\varepsilon/2}$.

Figure 4: **Inverse binary KL-divergence.** The figure shows plots of $kl^{-1}(q,\varepsilon)$ for $\varepsilon \in \{0, 0.1, 0.2, 0.5, 1, 2, 5\}$ in different colors, the curves for larger $\varepsilon$ lying higher. For $\varepsilon = 0$ it is $kl^{-1}(q,\varepsilon=0) = q$ (staight blue line). At $q = 0$ the curves start at $kl^{-1}(q=0,\varepsilon) = 1 - e^{-\varepsilon}$. At $q = 1$ we have $kl^{-1}(q=1,\varepsilon) = 1$ for any $\varepsilon \geq 0$.

When applying gradient descent on the RHS of the generalization bound (5) (which includes (4) as a special case), as we propose and do, one further needs, besides the evaluation of $kl^{-1}$, also the derivatives of $kl^{-1}$ w.r.t. both of its arguments. These can be easily derived by differentiating the identity $kl(q \parallel kl^{-1}(q,\varepsilon)) = \varepsilon$ w.r.t. $q$ and $\varepsilon$, plugging in the easily computed derivatives of $kl(q \parallel p) = q \ln \frac{q}{p} + (1-q) \ln \frac{1-q}{1-p}$. The result is:

$$\frac{\partial \, kl^{-1}(q,\varepsilon)}{\partial q} = \frac{\ln \frac{1-q}{1-kl^{-1}(q,\varepsilon)} - \ln \frac{q}{kl^{-1}(q,\varepsilon)}}{\frac{1-q}{1-kl^{-1}(q,\varepsilon)} - \frac{q}{kl^{-1}(q,\varepsilon)}}, \tag{12}$$

$$\frac{\partial \, kl^{-1}(q,\varepsilon)}{\partial \varepsilon} = \frac{1}{\frac{1-q}{1-kl^{-1}(q,\varepsilon)} - \frac{q}{kl^{-1}(q,\varepsilon)}}. \tag{13}$$

The derivative of the RHS of (5) with respect to parameters $\xi$ (which may include the hyperparameters $\theta$, the noise level $\sigma_n^2$, the inducing points $z_i$, or any other parameters of $P$ and $Q$ such as $a_M, B_{MM}$,

or $\alpha$ from Section 3.2) thus reads:

$$\frac{d}{d\xi} kl^{-1}\left(R_S(Q_\xi), \frac{KL(Q_\xi \| P_\xi) + \ln \frac{1}{p_\xi} + \ln \frac{2\sqrt{N}}{\delta}}{N}\right) =$$

$$= \left(\frac{\partial\, kl^{-1}(q,\varepsilon)}{\partial q}\Bigg|_{\substack{q=R_S(Q_\xi) \\ \varepsilon=(KL(Q_\xi \| P_\xi)+\ln\frac{2\sqrt{N}}{\delta p_\xi})/N}}\right) \cdot \frac{d}{d\xi} R_S(Q_\xi)$$

$$+ \left(\frac{\partial\, kl^{-1}(q,\varepsilon)}{\partial \varepsilon}\Bigg|_{\substack{q=R_S(Q_\xi) \\ \varepsilon=(KL(Q_\xi \| P_\xi)+\ln\frac{2\sqrt{N}}{\delta p_\xi})/N}}\right) \cdot \frac{d}{d\xi} \frac{KL(Q_\xi \| P_\xi) + \ln \frac{1}{p_\xi} + \ln \frac{2\sqrt{N}}{\delta}}{N},$$

$$(14)$$

using the partial derivatives of $kl^{-1}$ in parentheses from (12)–(13).

We use the expression (14) for our gradient-based optimization of the parameters $\xi$ in Sect. 4. Note that we treat all components of $\xi$ as continuous parameters during this optimization, despite the fact that the hyperparameters $\theta \in \Theta$ for the prior $P_\theta$ have to come from a *countable* set $\Theta$ (see around Eq. (5) and App. B). It is only after the optimization that we discretize all of those parameters $\theta$ onto a pre-defined grid (chosen before seeing the training sample $S$), as described in Sect. 3.1.

Note that $\frac{d}{d\xi} KL(Q_\xi \| P_\xi)$ in (14) can be computed analytically in our proposed methods PAC-GP and PAC-SGP by using standard matrix algebra (e.g., [1, App. A]) for differentiating the matrix-analytic expressions of $KL(Q_\xi \| P_\xi)$ in (7) or (10) (possibly with the parametrization (11)). Furthermore, the derivative $\frac{d}{d\xi} \ln \frac{1}{p_\xi} = -\frac{1}{p_\xi}\frac{d}{d\xi}p_\xi$ is easily computed for common distributions $p_\theta$ (Sect. 2.2), again treating $\xi$ first as a continuous parameter in the optimization as explained in the previous paragraph; in our paper, we always discretize the hyperparameter set $\Theta$ to be finite and choose $p_\xi = \frac{1}{|\Theta|}$ as the uniform distribution, so $p_\xi$ is independent of $\xi$ and $\frac{d}{d\xi} \ln \frac{1}{p_\xi} = 0$. Lastly, we show in App. C how to effectively compute $\frac{d}{d\xi} R_S(Q_\xi)$ in the expression (14) for relevant loss functions $\ell$.

To our knowledge, the parameter-free PAC-Bayes bound from Theorem 1 or Eq. (5) has never before been used for learning, as we do in our paper here, ostensibly due to the perceived difficulty of handling the derivatives of $kl^{-1}$ [14]. Instead, when a PAC-Bayes bound was used to guide learning in prior works [13, 14], then a simple sum of $R_S(Q)$ and a penalty term involving $KL(Q \| P)$ and $\log \frac{1}{p_\theta}$ was employed as an upper bound, either obtained from alternative PAC-Bayes theorems [9, 13] or from loosening the upper bound in Eq. (5) to an expression of the form $R_S(Q) + \sqrt{(KL(Q \| P) + \ln \frac{2\sqrt{N}}{\delta})/(2N)}$ by a use of Pinsker's inequality [14] or by looser derivations (some of which are mentioned in [13, 21]). We show in our work how to perform the learning directly with the $kl^{-1}$-bound (5) using the derivative from Eq. (14), and demonstrate that its optimization is robust and stable and has better performance than the optimization of looser bounds (see Sect. 4).

# B Proof of Eq. (5) — KL-divergence inequality and union bound

Let $\Theta$ be a *countable* set (i.e. a finite set or a countably infinite set), and let $p_\theta$ be any probability distribution over its elements $\theta \in \Theta$. Further, let $P^\theta$ be a family of probability distributions indexed by the $\theta \in \Theta$, and define their mixture $P := \sum_{\theta'} p_{\theta'} P^{\theta'}$. Then it holds for each $\theta \in \Theta$ and $Q$:

$$KL(Q \parallel P) = \int dx\, Q(x) \ln \frac{Q(x)}{P(x)} \tag{15}$$

$$= \int dx\, Q(x) \ln \frac{Q(x)}{\sum_{\theta'} p_{\theta'} P^{\theta'}(x)}$$

$$\leq \int dx\, Q(x) \ln \frac{Q(x)}{p_\theta P^\theta(x)} \tag{16}$$

$$= \int dx\, Q(x) \ln \frac{1}{p_\theta} \;+\; \int dx\, Q(x) \ln \frac{Q(x)}{P^\theta(x)}$$

$$= \ln \frac{1}{p_\theta} \;+\; KL(Q \parallel P^\theta). \tag{17}$$

The inequality (16) follows from the simple fact that the sum $\sum_{\theta'} p_{\theta'} P^{\theta'}(x)$ contains only nonnegative terms and is therefore at least as large as any of its summands, $\sum_{\theta'} p_{\theta'} P^{\theta'}(x) \geq p_\theta P^\theta(x)$, together with the monotonicity of the logarithm $\ln$. This inequality would not generally hold when $\sum_{\theta'}$ were replaced by an integral $\int_{\theta'}$ over a continuous index $\theta'$, which explains the requirement of a *countable* index set $\Theta$. The inequality $KL(Q \parallel P) \leq \ln \frac{1}{p_\theta} + KL(Q \parallel P^\theta)$ holds also for $p_\theta = 0$ with the interpretation $\ln \frac{1}{0} = \infty$.

The inequality $KL(Q \parallel P) \leq \ln \frac{1}{p_\theta} + KL(Q \parallel P^\theta)$ from (15)–(17), together with fact that $kl^{-1}$ is monotonically increasing in its second argument, shows how to obtain Eq. (5) from Theorem 1.

**Remark 2.** *Using the value $KL(Q \parallel P)$ with $P = \sum_{\theta'} p_{\theta'} P^{\theta'}$ directly in (4) would of course yield a better bound than (5), but this $KL(Q \parallel P)$ is generally difficult to evaluate, e.g. when $P$ is a mixture of Gaussians. Furthermore, the alternative bound $KL(Q \parallel \sum_{\theta'} p_{\theta'} P^{\theta'}) \leq \sum_{\theta'} p_{\theta'} KL(Q \parallel P^{\theta'})$, coming from convexity of $KL$, would require the value of $KL(Q \parallel P^{\theta'})$ for each $\theta' \in \Theta$; but $KL(Q \parallel P^{\theta'})$ cannot be computed by the automatic method of Sections 3.1–3.2 when $Q$ originates from different hyperparameters than $P^{\theta'}$.*

As an alternative derivation of Eq. (5) from Theorem 1, one may use the ordinary union bound argument: For a given probability distribution $p_\theta$ on the countable set $\Theta$ and a given $\delta \in (0, 1]$, define $\delta_\theta := \delta p_\theta$. Now consider the statement of Theorem 1 for each prior $P^\theta$ individually with confidence parameter $\delta_\theta$; this gives that, for each $\theta \in \Theta$, the statement

$$\forall Q: \quad R(Q) \leq kl^{-1}\left(R_S(Q), \frac{KL(Q \parallel P^\theta) + \ln \frac{2\sqrt{N}}{\delta_\theta}}{N}\right)$$

$$= kl^{-1}\left(R_S(Q), \frac{KL(Q \parallel P^\theta) + \ln \frac{1}{p_\theta} + \ln \frac{2\sqrt{N}}{\delta}}{N}\right)$$

fails with probability at most $\delta_\theta$ (over $S \sim \mu^N$). By the union bound, the statement fails for one $\theta \in \Theta$ with probability at most $\sum_\theta \delta_\theta = \sum_\theta \delta p_\theta = \delta \sum_\theta p_\theta = \delta \cdot 1 = \delta$. Thus, the statement of Eq. (5) (containing the quantifier $\forall \theta$) holds with probability at least $1 - \delta$ over $S \sim \mu^N$.

Figure 5: **Various regression loss functions.** Shown are three bounded loss functions $\ell$, which are appropriate for the regression setting and which allow for an effective computation of the empirical risk $R_S(Q)$ when $Q$ is a GP. Each of these three functions $\ell(y, \widehat{y})$ depends only on the absolute deviation $y - \widehat{y}$ (horizontal axis), and contains a scale parameter $\varepsilon > 0$ which is set to $\varepsilon = 1$ in the plots: $\ell_{\mathbb{1}}(y, \widehat{y}) = \mathbb{1}_{|y-\widehat{y}|>\varepsilon} = \mathbb{1}_{\widehat{y} \notin [y-\varepsilon, y+\varepsilon]}$ (blue), which we use in our experiments, $\ell_2(y, \widehat{y}) = \min\{[(y - \widehat{y})/\varepsilon]^2, 1\}$ (red), and $\ell_{\exp}(y, \widehat{y}) = 1 - \exp[-((y - \widehat{y})/\varepsilon)^2]$ (yellow).

## C  Loss functions, the empirical risk $R_S(Q)$, and its gradient

Our proposed method requires the empirical risk $R_S(Q)$ on the training set $S = \{(x_i, y_i)\}_{i=1}^N$ (see Sect. 2.1) to be computed effectively for any considered distribution $Q$, along with its gradient $\frac{d}{d\xi} R_S(Q_\xi)$ for gradient-based optimization. We show here that this can be done for many interesting loss functions $\ell$ when $Q$ is a Gaussian Process, including the following (see Fig. 5 for illustration):

$$\ell_{\mathbb{1}}(y, \widehat{y}) = \mathbb{1}_{|y-\widehat{y}|>\varepsilon} = \mathbb{1}_{\widehat{y} \notin [y-\varepsilon, y+\varepsilon]}, \tag{18}$$

$$\ell_2(y, \widehat{y}) = \min\{((y - \widehat{y})/\varepsilon)^2, 1\}, \tag{19}$$

$$\ell_{\exp}(y, \widehat{y}) = 1 - \exp[-((y - \widehat{y})/\varepsilon)^2], \tag{20}$$

$$\ell_\pm(y, \widehat{y}) = \mathbb{1}_{\widehat{y} \notin [r_-(y), r_+(y)]}, \tag{21}$$

where $\varepsilon > 0$ is a scale parameter to be chosen for the first three, and $r_\pm(y)$ are functions to be specified for the last. Note that $\ell_{\mathbb{1}}$ specifies an *additive* accuracy goal $\pm\varepsilon$ and was used in our experiments (Sect. 4), whereas we have suggested $\ell_2$ and $\ell_{\exp}$ as more deviation-sensitive (yet bounded) loss functions that may yield better results on the MSE error (see Sect. 5, and Sect. 4). The loss function $\ell_\pm$ generalizes $\ell_{\mathbb{1}}$ (which is obtained by using the functions $r_\pm(y) := y \pm \varepsilon$, see Sect. 2.1), but could also be used to specify *relative* accuracy goals, e.g. setting $r_\pm(y) := y \pm \varepsilon|y|$. More deviation-sensitive relative loss functions are possible as well, e.g. $\ell(y, \widehat{y}) := \max\left\{\left|\frac{\widehat{y}-y}{y}\right|, 1\right\}$, which we do not treat here but which allows similarly effective computation as the other ones.

Let us denote by $\widehat{m}(x)$ and $\widehat{\sigma}^2(x)$ the predictive mean and variance of the predictive GP $Q$. In our work we use the two forms (6) (PAC-GP) and (9) (sparse PAC-SGP); in the latter case we e.g. have:

$$\widehat{m}(x) = m(x) + k_M(x)K_{MM}^{-1}(a_M - m_M), \tag{22}$$

$$\widehat{\sigma}^2(x) = K(x, x') - k_M(x)K_{MM}^{-1}[K_{MM} - B_{MM}]K_{MM}^{-1}k_M(x')^T. \tag{23}$$

We denote by $\widehat{m}_i := \widehat{m}(x_i)$, $\widehat{\sigma}_i^2 := \widehat{\sigma}^2(x_i)$ the predictive mean and variance at the training inputs. The empirical risk $R_S(Q)$ from (1) then reduces to a sum of one-dimensional integrals containing a Gaussian density:

$$R_S(Q) = \frac{1}{N} \sum_{i=1}^N \mathbb{E}_{h \sim Q}\big[\ell\big(y_i, h(x_i)\big)\big] = \frac{1}{N} \sum_{i=1}^N \mathbb{E}_{v \sim Q(x_i)}\big[\ell\big(y_i, v\big)\big] \tag{24}$$

$$= \frac{1}{N} \sum_{i=1}^N \int dv \, \mathcal{N}(v \mid \widehat{m}_i, \widehat{\sigma}_i^2) \, \ell(y_i, v). \tag{25}$$

The last integral can be evaluated for each of the loss functions (18)–(21):

$$\int dv\, \mathcal{N}(v \mid \widehat{m}_i, \widehat{\sigma}_i^2)\, \ell_{\mathbb{1}}(y_i, v) = \Phi\left(\frac{y_i - \varepsilon - \widehat{m}_i}{\widehat{\sigma}_i}\right) + 1 - \Phi\left(\frac{y_i + \varepsilon - \widehat{m}_i}{\widehat{\sigma}_i}\right), \tag{26}$$

$$\int dv\, \mathcal{N}(v \mid \widehat{m}_i, \widehat{\sigma}_i^2)\, \ell_2(y_i, v) = \left(1 - \frac{(y_i - \widehat{m}_i)^2 + \widehat{\sigma}_i^2}{\varepsilon^2}\right)\left(\Phi\left(\frac{y_i - \varepsilon - \widehat{m}_i}{\widehat{\sigma}_i}\right) - \Phi\left(\frac{y_i + \varepsilon - \widehat{m}_i}{\widehat{\sigma}_i}\right)\right)$$
$$+ 1 - \frac{\widehat{\sigma}_i}{\sqrt{2\pi}\varepsilon^2}(y_i - \varepsilon - \widehat{m}_i)e^{-(y_i + \varepsilon - \widehat{m}_i)^2/(2\widehat{\sigma}_i^2)} \tag{27}$$
$$- \frac{\widehat{\sigma}_i}{\sqrt{2\pi}\varepsilon^2}(y_i + \varepsilon - \widehat{m}_i)e^{-(y_i - \varepsilon - \widehat{m}_i)^2/(2\widehat{\sigma}_i^2)},$$

$$\int dv\, \mathcal{N}(v \mid \widehat{m}_i, \widehat{\sigma}_i^2)\, \ell_{\exp}(y_i, v) = 1 - \frac{1}{\sqrt{1 + \frac{2\widehat{\sigma}_i^2}{\varepsilon^2}}} \exp\left[-\frac{(y_i - \widehat{m}_i)^2}{2\widehat{\sigma}_i + \varepsilon^2}\right], \tag{28}$$

$$\int dv\, \mathcal{N}(v \mid \widehat{m}_i, \widehat{\sigma}_i^2)\, \ell_{\pm}(y_i, v) = \Phi\left(\frac{r_-(y_i) - \widehat{m}_i}{\widehat{\sigma}_i}\right) + 1 - \Phi\left(\frac{r_+(y_i) - \widehat{m}_i}{\widehat{\sigma}_i}\right), \tag{29}$$

where by

$$\Phi(z) := \int_{-\infty}^{z} \frac{1}{\sqrt{2\pi}} e^{-t^2/2} dt \tag{30}$$

we denote the cumulative distribution function of a standard normal, which is implemented in most computational packages. Plugging the expressions (26)–(29) into (25) shows how $R_S(Q)$ can be computed.

With the above expressions one can also compute gradients of $R_S(Q) = R_S(Q_\xi)$ effectively for gradient-based optimization: When $Q = Q_\xi$ depends on parameters $\xi$ (such as hyperparameters $\theta$, noise $\sigma_n$, inducing points $\{z_i\}$, or any other free-form parameters $a_M$, $B_{MM}$ or $\alpha$ from Sect. 3.2), then $\widehat{m}(x) = \widehat{m}^\xi(x)$ and $\widehat{\sigma}(x) = \widehat{\sigma}^\xi(x)$ depend on $\xi$ as well through explicit expressions, via (6) and (9). One can thus compute the gradients $\frac{d}{d\xi}\widehat{m}_i^\xi = \frac{d}{d\xi}\widehat{m}^\xi(x_i)$ and $\frac{d}{d\xi}\widehat{\sigma}_i^\xi = \frac{d}{d\xi}\widehat{\sigma}^\xi(x_i)$ analytically, using standard matrix analysis (e.g. [1, App. A]). With these gradients and the above expressions (26)–(29) it is easy to compute $\frac{d}{d\xi}R_S(Q_\xi)$ for the above loss function; e.g. for $\ell_{\mathbb{1}}$ from (18) used in our experiments:

$$\frac{d}{d\xi}R_S^{\mathbb{1}}(Q_\xi) = \frac{1}{N}\sum_{i=1}^{N}\left[\left(\frac{d}{d\xi}\frac{y_i - \varepsilon - \widehat{m}_i}{\widehat{\sigma}_i}\right)e^{-\frac{1}{2}\left(\frac{y_i - \varepsilon - \widehat{m}_i}{\widehat{\sigma}_i}\right)^2} - \left(\frac{d}{d\xi}\frac{y_i + \varepsilon - \widehat{m}_i}{\widehat{\sigma}_i}\right)e^{-\frac{1}{2}\left(\frac{y_i + \varepsilon - \widehat{m}_i}{\widehat{\sigma}_i}\right)^2}\right] \tag{31}$$

where we used that $\frac{d}{dz}\Phi(z) = \frac{1}{\sqrt{2\pi}}e^{-z^2/2}$.

Lastly, for the purpose of gradient-based optimization of the objective from Theorem 1 or Eq. (5), one does not really need to compute the exact $R_S(Q)$ as a sum over $N$ training examples, which is possibly a large number. Rather, one could do mini-batches of size $B \ll N$ and obtain a stochastic estimate

$$R_S(Q) \approx \frac{1}{B}\sum_{i=1}^{B}\int dv\, \mathcal{N}(v \mid \widehat{m}_i, \widehat{\sigma}_i^2)\, \ell(y_i, v) =: \widehat{R}_B(Q), \tag{32}$$

where the sum runs over one mini-batch selected from the $N$ training points randomly or in cyclic order. (Hoeffding's inequality gives that $|R_S(Q) - \widehat{R}_B(Q)| \lesssim \sqrt{\frac{1}{2B}\ln\frac{2}{\delta'}}$ holds with probability $\geq 1 - \delta'$ over mini-batches. While this statement could be incorporated into a version of Theorem 1 or Eq. (5) that is expressed in terms of $\widehat{R}_B(Q)$ instead of $R_S(Q)$, we propose stochastic estimates $\widehat{R}_B(Q)$ only during the optimization procedure and suggest a full computation of $R_S(Q)$ for the final evaluation of the generalization bound.) Similarly, the exact gradient $\frac{d}{d\xi}R_S(Q_\xi)$ is a sum over $N$ training examples (e.g., (31)), so one can approximate it in the same way by mini-batches to obtain a faster stochastic estimate of the gradient which is often sufficient for optimization.

## D  Training objectives of other GP methods

Here we contrast our proposed learning objective (5) with those of other common GP methods, to which we compare in the experiments (Sect. 4).

In standard full GP regression learning [1] one selects those prior hyperparameters $\theta$ and noise level $\sigma_n$ which maximize the data likelihood $p(y_N \mid \theta, \sigma_n) = \mathcal{N}(y_N \mid m_N, K_{NN} + \sigma_n^2 \mathbb{1})$ under the *prior* GP. This corresponds to the minimization objective

$$-\ln p(y_N \mid \theta, \sigma_n) = \frac{1}{2} \ln \det[K_{NN} + \sigma_n^2 \mathbb{1}] + \frac{N}{2} \ln(2\pi) + \frac{1}{2}(y_N - m_N)^T (K_{NN} + \sigma_n^2 \mathbb{1})^{-1}(y_N - m_N). \tag{33}$$

The optimal $\theta$, $\sigma_n$ are then used in (6) to make predictions.

The sparse-GP methods FITC [15], VFE [6], and DTC [16] adjust $\theta$, $\sigma_n$, and the $M$ inducing inputs $\{z_i\}$ by minimizing the objective [2]

$$\mathcal{F} = \frac{1}{2} \ln \det \left[ K_{NM} K_{MM}^{-1} K_{MN} + \sigma_n^2 \mathbb{1} + G \right] + \frac{N}{2} \ln(2\pi) + \frac{1}{2\sigma_n^2} \text{tr}[T]$$
$$+ \frac{1}{2}(y_N - m_N)^T \left( K_{NM} K_{MM}^{-1} K_{MN} + \sigma_n^2 \mathbb{1} + G^\theta \right)^{-1}(y_N - m_N), \tag{34}$$

where $G_{\text{FITC}} = T_{VFE} = \text{diag}(K_{NN} - K_{NM} K_{MM}^{-1} K_{MN})$ and $G_{\text{VFE}} = G_{\text{DTC}} = T_{FITC} = T_{DTC} = 0$. For DTC and FITC, $\mathcal{F}$ are the negative log likelihoods of approximate prior models [24, 15], whereas $\mathcal{F}$ equals the exact negative log likelihood plus the KL-divergence $KL(Q \parallel \widetilde{Q})$ between $Q$ and the exact Bayesian posterior $\widetilde{Q}$ obtained from the Bayesian prior $P$. VFE and DTC make predictions $Q$ by using (11) with $\alpha = 0$ in (9), whereas FITC sets $\alpha = 1$.

One can compare the above expressions to the $KL(Q \parallel P)$ term in the PAC-Bayes bound (5). For our full-GP training, $KL(Q \parallel P)$ is given in (8):

$$KL(Q \parallel P) = -\frac{1}{2} \ln \det \left[ K_{NN}^{-1}(K_{NN} - K_{NN}(K_{NN} + \sigma_n^2 \mathbb{1})^{-1} K_{NN}) \right]$$
$$+ \frac{1}{2} \text{tr}[K_{NN}^{-1}(K_{NN} - K_{NN}(K_{NN} + \sigma_n^2 \mathbb{1})^{-1} K_{NN})] - \frac{N}{2}$$
$$+ \frac{1}{2}(y_N - m_N)^T (K_{NN} + \sigma_n^2 \mathbb{1})^{-1} K_{NN}(K_{NN} + \sigma_n^2 \mathbb{1})^{-1}(y_N - m_N),$$
$$= \frac{1}{2} \ln \det \left[ K_{NN} + \sigma_n^2 \mathbb{1} \right] - \frac{N}{2} \ln \sigma_n^2 - \frac{1}{2} \text{tr} \left[ K_{NN}(K_{NN} + \sigma_n^2 \mathbb{1})^{-1} \right]$$
$$+ \frac{1}{2}(y_N - m_N)^T (K_{NN} + \sigma_n^2 \mathbb{1})^{-1} K_{NN}(K_{NN} + \sigma_n^2 \mathbb{1})^{-1}(y_N - m_N),$$
$$= \frac{1}{2} \sum_{i=1}^{N} \left[ \ln \frac{\lambda_i + \sigma_n^2}{\sigma_n^2} - \frac{\lambda_i}{\lambda_i + \sigma_n^2} \right] + \frac{1}{2} \sum_{i=1}^{N} \frac{\lambda_i}{(\lambda_i + \sigma_n^2)^2} (e_i \cdot (y - m_N))^2,$$

where $\lambda_i \in \mathbb{R}$ are the eigenvalues of $K_{NN}$ and $e_i \in \mathbb{R}^N$ corresponding orthonormal eigenvectors. For our sparse-GP training with a "free-form" sparsification $Q(f_M) = \mathcal{N}(f_M \mid a_M, B_{MM})$ with free $a_M$, $B_{MM}$, it is from (10):

$$KL(Q \parallel P) = KL(Q(f_M) \parallel P(f_M)) = -\frac{1}{2} \ln \det \left[ B_{MM} K_{MM}^{-1} \right] + \frac{1}{2} \text{tr} \left[ B_{MM} K_{MM}^{-1} \right] - \frac{M}{2}$$
$$+ \frac{1}{2}(a_M - m_M)^T K_{MM}^{-1}(a_M - m_M),$$

which via $a_M = K_{MM} Q_{MM}^{-1} K_{MN}(\alpha \Lambda + \sigma_n^2 \mathbb{1})^{-1} y_N$, $B_{MM} = K_{MM} Q_{MM}^{-1} K_{MM}$ from (11) with $\alpha = 1$ can be particularized for the FITC parametrization used in our PAC-SGP work.

# E Experiment: predictive distributions of sparse GPs, and overfitting

To compare the predictive distributions of common sparse GPs to the predictive distribution obtained from our sparse PAC-SGP method (Sect. 3.2) optimized with the PAC-Bayesian bound (5), we trained FITC [15] and VFE [6] on the same dataset used in Fig. 1, which was also used in [15, 6] for a comparison of methods. It can be seen in Fig. 6 that especially for small $\varepsilon$ our PAC-SGP has a predictive distribution more similar to FITC, whereas for larger $\varepsilon$, the predictive distribution becomes closer to the full-GP, however not as close as VFE. Note that, for the full-GP, for FITC and for VFE we include the additive observation noise $\sigma_n^2$ in the predictive uncertainty in Fig. 6, whereas for our PAC-SGP variant we do not include additive observation noise $\sigma_n^2$, since this is not part of the predictive variance (see Eqs. (9,11), and similarly Eq. (6) for the non-sparse case); we instead plot the $\varepsilon$-band from the 0-1-loss function (green) around the predictive PAC-SGP mean. We further refer to the discussions in [15, 6] concerning the same dataset.

Figure 6: **Comparison of predictive distributions.** In each plot, we show the predictive distribution from a full-GP in red (mean $\pm$ twice the predictive variance), fitted to the data (black dots). The blue distributions (mean $\pm$ twice the predictive variance) in the first two plots are obtained form our sparse kl-PAC-SGP with two different values of $\varepsilon$ (chosen relative to the noise level $\sigma_n = 0.28$ of the full-GP), the third shows the predictive distribution from FITC, the fourth from VFE. For the PAC-GP variants, we additionally plotted the $\varepsilon$-band as in Fig. 1. As in Fig. 1, the crosses show the inducing point positions before and after training.

As a further comparison of our method with FITC, we now illustrate the well-known overfitting of the FITC method on pathological datasets [2] and show how our PAC-SGP method avoids it. The dataset for this demonstration consists of half of the datapoints (using every second one) of the above 1D-dataset [15, 6], similar to what was done in the comparison study in [2, Section 3.1]. For 100 different initializations of $\sigma_n^2 \in [10^{-5}, 10^{+1}]$ and the $M = 8$ inducing inputs, we trained a FITC model and a kl-PAC-SGP model, minimizing the (approx.) negative log-likelihood for FITC and minimizing the BKL bound from Eq. (5) for kl-PAC-SGP (using the 0-1-loss function with $\varepsilon = 0.6$, cf. Sect. 4). Fig. 7 shows, for each of the (local) optima reached in these optimizations, the optimal learned noise variance $\sigma_n^2$ and the obtained values of the objective function at each local minimum.

For FITC, the learned noise variances $\sigma_n^2$ span five orders of magnitude, and many of them have very small values $\sim 10^{-6}$, lying outside of the initialization interval, and clearly overfit on the data (see [2, Figure 1]). Worse than that, the global optimum for FITC (red dot in left panel of Fig. 7) is found at the very small value of $\sigma_n^2 \sim 10^{-6}$, reproducing the findings of [2, Section 3.1]. In contrast to that, our kl-PAC-SGP is much better behaved: the local optima have more reasonable $\sigma_n^2 \in [2 \cdot 10^{-3}, 10^{-1}]$ and our global optimum has $\sigma_n^2 \approx 2.1 \cdot 10^{-2}$ (note however that the values of $\sigma_n$ learned by PAC-(S)GP will depend on the lengthscale $\varepsilon$ chosen for the loss function $\ell$; see also Table 1 in App. G). While kl-PAC-SGP has further local optima at the small values $\sigma_n^2 \in [10^{-5}, 2 \cdot 10^{-3}]$, where $\sigma_n^2$ does not move away from its small initialization value, these are easy to detect as the minimization objective attains the trivial value of $\approx 1$.

This shows that our PAC-GP method is more stable than FITC on these pathological datasets and returns a more reasonable estimate of the noise level $\sigma_n^2$. It also reinforces the finding from the experiments in Sect. 4 that our PAC-GP tends to underfit rather than overfit, hedging against violations of Theorem 1 and Eq. (5) by returning predictive GPs $Q$ of lower complexity $KL(Q\|P)$ by choosing larger $\sigma_n^2$.

Figure 7: **Local minima of the optimization for different initializations.** Shown are the learned $\sigma_n^2$ and the achieved (local) minima for 100 different initializations of $\sigma_n^2 \in [10^{-5}, 10^{+1}]$ for the FITC and kl-PAC-SGP methods trained on 100 out of the 200 datapoints of the 1D-dataset from [15, 6]. See also [2, Section 3.1].

## F  Experiment: dependence of the upper bound on discretization

In order to assess the effect of discretizing hyperparameters $\theta$ (see Sections 3.1 and 4) on the performance of the resulting GP and on the upper bound, we ran our PAC-GP from Sect. 3.1 with different discretization settings and fitted them to artificial data. The results are shown in Fig. 8.

Specifically, we generated inputs by uniformly sampling $x \in X := [-3, 3]^3 \subset \mathbb{R}^3$. We sampled $N = 2000$ training and $N = 10000$ test outputs by sampling from a GP on the generated inputs, using an SE-ARD-kernel with randomly selected lengthscales for each of the $d = 3$ dimensions. In more details, we sampled the kernel's log-lengthscales uniformly between $-1$ and $1$. To the generated data, we fitted a PAC-GP with a discretization given by $L \in \{1, 2, 4, 8\}$ (see Sect. 3.1) and a number of rounding digits $r \in \{0, 1, 2, 4\}$ (i.e. $G = 2L \cdot 10^r$ in Sect. 3.1). For example, for $L = 1, r = 0$, we only consider values $\log \theta \in \{-1, 0, 1\}$ resulting in $\log |\Theta| = (d+1) \cdot \ln(G+1) = 4 \ln 3 \approx 4.4$.

To assess the contribution of the training risk $R_S$ and the $KL$-divergence term to the overall upper bound (5) on the generalization performance, we plotted the mean of each of these terms as a function of $\log |\Theta|$, averaged across 68 repetitions. Additionally, we plotted the risk on a test set to assess whether the actual test performance $R(Q)$ is affected by coarser discretization of the GP hyperparameters. It can be seen in Fig. 8 that, as long as a minimal discrimination ability is allowed, both the training as well as the test risks are not affected by discretizing to a coarse grid of hyperparameters. Specifically, the jump that can be observed at $\ln |\Theta| \sim 11.3$ corresponds to going from $r = 0$ to $r \geq 1$, thereby keeping at least one decimal place in the discretization. We see that both the KL-divergence $KL(Q\|P)$ as well as the training risk $R_S(Q)$ is basically unaffected by the discretization for $r \geq 1$, so any increase in the resulting upper bound is due to the increase in $\log |\Theta|$.

From this investigation, we find the discretization with $L = 6$ and $r = 2$ to be completely sufficient for accuracy, while the resulting $\ln |\Theta|$ term is still small compared to the contribution $KL(Q\|P)$ in the PAC-Bayes bound (5) as seen in our experiments (Sect. 4). For this discretization we have $\ln |\Theta| = (d+1) \ln(1201) \approx 7.1(d+1)$ for an SE-ARD kernel in $d$ dimensions, and $\ln |\Theta| = 2 \ln(1201) \approx 14.2$ for a non-ARD SE-kernel. Note that – for any fixed rounding accuracy of $\sim \log_2 G$ bits – the penalty term $\ln |\Theta|$ as well as the required storage capacity and computational effort all scale only linearly with the input dimension $d$; thus, our method requires the same computational complexity as other standard GP methods.

Figure 8: **Analysis of the discretization effect.** Upper bound (5) and its contributing terms, as well as the training and test risks, as a function of the discretization as measured by $\log|\Theta|$. Each line corresponds to the mean value over 68 iterations, when trained with our PAC-GP fitted to 3-dimensional data generated from an SE-ARD kernel with random lengthscales (see text, App. F).

# G  Supplementary Tables

Table 1: **Evaluation of full GP models (Fig. 2).** We compare our approach ("kl-PAC-GP") with minimizing the looser bound $B_{Pin}$ ("sqrt-PAC GP") and with the standard GP approach ("full-GP" [1]) on the following metrics (from left to right): upper bound $B$, Pinsker's upper bound $B_{Pin}$, Gibbs risk on the training data $R_S$[train], Gibbs risk on the test data $R_S$[test], mean squared error (MSE) on the test data, KL-divergence (normalized by the number of training samples, i.e. $KL(Q\|P)/N$), and the learned noise parameter $\sigma_n^2$. Shown are the averages± standard errors over 10 repetitions.

| Model configuration | | | Upper bound | | Gibbs risk | | MSE | Model properties | |
| --- | --- | --- | --- | --- | --- | --- | --- | --- | --- |
| dataset | epsilon | method | $B$ | $B_{Pin}$ | $R_S$[train] | $R_S$[test] | MSE | KL/N | $\sigma^2$ |
| boston | 0.2 | kl-PAC-GP | 0.773 +/- 0.003 | 0.798 +/- 0.003 | 0.497 +/- 0.005 | 0.536 +/- 0.006 | 0.159 +/- 0.018 | 0.126 +/- 0.003 | 0.304 +/- 0.015 |
| | | sqrt-PAC-GP | 0.803 +/- 0.016 | 0.834 +/- 0.019 | 0.573 +/- 0.039 | 0.599 +/- 0.034 | 0.420 +/- 0.129 | 0.087 +/- 0.018 | 1334.991 +/- 1334.032‡‡ |
| | | full-GP | 0.809 +/- 0.004 | 0.851 +/- 0.006 | 0.372 +/- 0.012 | 0.501 +/- 0.007 | 0.114 +/- 0.015 | 0.405 +/- 0.023 | 0.066 +/- 0.004 |
| | 0.4 | kl-PAC-GP | 0.498 +/- 0.004 | 0.507 +/- 0.003 | 0.211 +/- 0.005 | 0.243 +/- 0.008 | 0.161 +/- 0.018 | 0.120 +/- 0.002 | 0.328 +/- 0.013 |
| | | sqrt-PAC-GP | 0.498 +/- 0.004 | 0.507 +/- 0.003 | 0.218 +/- 0.004 | 0.250 +/- 0.008 | 0.165 +/- 0.019 | 0.111 +/- 0.002 | 0.371 +/- 0.013 |
| | | full-GP | 0.548 +/- 0.005 | 0.576 +/- 0.006 | 0.097 +/- 0.008 | 0.217 +/- 0.007 | 0.114 +/- 0.015 | 0.405 +/- 0.023 | 0.066 +/- 0.004 |
| | 0.6 | kl-PAC-GP | 0.333 +/- 0.004 | 0.376 +/- 0.002 | 0.093 +/- 0.003 | 0.115 +/- 0.006 | 0.167 +/- 0.019 | 0.104 +/- 0.002 | 0.424 +/- 0.015 |
| | | sqrt-PAC-GP | 0.336 +/- 0.003 | 0.373 +/- 0.002 | 0.111 +/- 0.003 | 0.133 +/- 0.006 | 0.182 +/- 0.020 | 0.082 +/- 0.001 | 0.625 +/- 0.014 |
| | | full-GP | 0.432 +/- 0.009 | 0.503 +/- 0.010 | 0.025 +/- 0.003 | 0.096 +/- 0.008 | 0.114 +/- 0.015 | 0.405 +/- 0.023 | 0.066 +/- 0.004 |
| | 0.8 | kl-PAC-GP | 0.247 +/- 0.003 | 0.313 +/- 0.002 | 0.053 +/- 0.002 | 0.069 +/- 0.005 | 0.181 +/- 0.019 | 0.080 +/- 0.002 | 0.666 +/- 0.017 |
| | | sqrt-PAC-GP | 0.253 +/- 0.003 | 0.308 +/- 0.002 | 0.072 +/- 0.002 | 0.089 +/- 0.006 | 0.212 +/- 0.022 | 0.055 +/- 0.001 | 1.163 +/- 0.020 |
| | | full-GP | 0.394 +/- 0.011 | 0.486 +/- 0.011 | 0.008 +/- 0.001 | 0.046 +/- 0.006 | 0.114 +/- 0.015 | 0.405 +/- 0.023 | 0.066 +/- 0.004 |
| | 1.0 | kl-PAC-GP | 0.198 +/- 0.002 | 0.278 +/- 0.001 | 0.035 +/- 0.002 | 0.047 +/- 0.004 | 0.196 +/- 0.020 | 0.062 +/- 0.001 | 1.025 +/- 0.028 |
| | | sqrt-PAC-GP | 0.206 +/- 0.002 | 0.271 +/- 0.001 | 0.052 +/- 0.001 | 0.066 +/- 0.005 | 0.238 +/- 0.023 | 0.040 +/- 0.001 | 1.987 +/- 0.036 |
| | | full-GP | 0.379 +/- 0.013 | 0.481 +/- 0.012 | 0.003 +/- 0.000 | 0.026 +/- 0.004 | 0.114 +/- 0.015 | 0.405 +/- 0.023 | 0.066 +/- 0.004 |

Table 2: **Evaluation of sparse GP models (Fig. 3)**. We benchmark our method ("kl-PAC-SGP") against minimizing the looser bound $B_{Pin}$ ("sqrt-PAC-SGP") and two standard sparse GP approaches (VFE [6] and VFE [15]) using the following criteria (from left to right): upper bound $B$, Pinsker's upper bound $B_{Pin}$, Gibbs risk on the training data $R_S$[train], Gibbs risk on the test data $R_S$[test], mean squared error (MSE) on the test data, KL-divergence (normalized by the number of training samples, i.e. $KL(Q\|P)/N$), and the learned noise parameter $\sigma_n^2$. The number of inducing inputs is fixed to $M = 500$, and we use the 0-1-loss function $\ell(\hat{y}, \tilde{y}) = \mathbb{1}_{\tilde{y} \notin [y-\varepsilon, y+\varepsilon]}$ with $\varepsilon = 0.6$ (see Sect. 4). Automatic feature determination (ARD) is beneficial on the datasets *pol* and *kin40k* and has no effect on *sarcos*. We report mean values ± standard errors over 10 iterations.

| Model configuration | | | Upper bound | | Gibbs risk | | | Model properties | |
| --- | --- | --- | --- | --- | --- | --- | --- | --- | --- |
| dataset | method | ARD | $B$ | $B_{Pin}$ | $R_S$[train] | $R_S$[test] | MSE | KL/N | $\sigma^2$ |
| pol | kl-PAC-SGP | ✗ | 0.217 +/- 0.001 | 0.252 +/- 0.000 | 0.106 +/- 0.001 | 0.115 +/- 0.001 | 0.114 +/- 0.001 | 0.041 +/- 0.000 | 0.316 +/- 0.015 |
| | sqrt-PAC-SGP | ✗ | 0.221 +/- 0.001 | 0.248 +/- 0.000 | 0.126 +/- 0.001 | 0.133 +/- 0.001 | 0.124 +/- 0.001 | 0.028 +/- 0.000 | 0.626 +/- 0.025 |
| | VFE | ✗ | 0.257 +/- 0.000 | 0.312 +/- 0.000 | 0.071 +/- 0.000 | 0.081 +/- 0.001 | 0.090 +/- 0.001 | 0.114 +/- 0.000 | 0.102 +/- 0.000 |
| | FITC | ✗ | 0.359 +/- 0.001 | 0.384 +/- 0.001 | 0.149 +/- 0.001 | 0.160 +/- 0.001 | 0.092 +/- 0.001 | 0.109 +/- 0.002 | 0.000 +/- 0.000 |
| | kl-PAC-SGP | ✓ | 0.083 +/- 0.000 | 0.172 +/- 0.000 | 0.011 +/- 0.000 | 0.015 +/- 0.000 | 0.036 +/- 0.000 | 0.035 +/- 0.000 | 0.187 +/- 0.003 |
| | sqrt-PAC-SGP | ✓ | 0.094 +/- 0.000 | 0.159 +/- 0.000 | 0.029 +/- 0.000 | 0.032 +/- 0.000 | 0.044 +/- 0.000 | 0.017 +/- 0.000 | 0.825 +/- 0.011 |
| | VFE | ✓ | 0.198 +/- 0.000 | 0.324 +/- 0.000 | 0.002 +/- 0.000 | 0.006 +/- 0.000 | 0.015 +/- 0.000 | 0.190 +/- 0.000 | 0.016 +/- 0.000 |
| | FITC | ✓ | 0.247 +/- 0.001 | 0.333 +/- 0.001 | 0.029 +/- 0.000 | 0.032 +/- 0.001 | 0.027 +/- 0.000 | 0.168 +/- 0.001 | 0.000 +/- 0.000 |
| sarcos | kl-PAC-SGP | ✗ | 0.031 +/- 0.000 | 0.083 +/- 0.000 | 0.009 +/- 0.000 | 0.010 +/- 0.000 | 0.033 +/- 0.000 | 0.010 +/- 0.000 | 0.526 +/- 0.004 |
| | sqrt-PAC-SGP | ✗ | 0.038 +/- 0.000 | 0.066 +/- 0.000 | 0.023 +/- 0.000 | 0.023 +/- 0.000 | 0.044 +/- 0.000 | 0.003 +/- 0.000 | 3.600 +/- 0.006 |
| | VFE | ✗ | 0.097 +/- 0.000 | 0.215 +/- 0.000 | 0.002 +/- 0.000 | 0.003 +/- 0.000 | 0.017 +/- 0.000 | 0.090 +/- 0.000 | 0.019 +/- 0.000 |
| | FITC | ✗ | 0.116 +/- 0.000 | 0.211 +/- 0.000 | 0.014 +/- 0.000 | 0.015 +/- 0.000 | 0.019 +/- 0.000 | 0.076 +/- 0.000 | 0.000 +/- 0.000 |
| | kl-PAC-SGP | ✓ | 0.031 +/- 0.000 | 0.095 +/- 0.000 | 0.005 +/- 0.000 | 0.007 +/- 0.000 | 0.029 +/- 0.000 | 0.012 +/- 0.000 | 0.389 +/- 0.002 |
| | sqrt-PAC-SGP | ✓ | 0.039 +/- 0.000 | 0.079 +/- 0.000 | 0.018 +/- 0.000 | 0.018 +/- 0.000 | 0.040 +/- 0.000 | 0.003 +/- 0.000 | 2.682 +/- 0.009 |
| | VFE | ✓ | 0.092 +/- 0.000 | 0.212 +/- 0.000 | 0.002 +/- 0.000 | 0.002 +/- 0.000 | 0.016 +/- 0.000 | 0.084 +/- 0.000 | 0.017 +/- 0.000 |
| | FITC | ✓ | 0.115 +/- 0.000 | 0.215 +/- 0.000 | 0.012 +/- 0.000 | 0.012 +/- 0.000 | 0.017 +/- 0.000 | 0.079 +/- 0.000 | 0.000 +/- 0.000 |
| kin40k | kl-PAC-SGP | ✗ | 0.154 +/- 0.000 | 0.219 +/- 0.000 | 0.045 +/- 0.000 | 0.053 +/- 0.000 | 0.059 +/- 0.001 | 0.059 +/- 0.000 | 0.262 +/- 0.014 |
| | sqrt-PAC-SGP | ✗ | 0.162 +/- 0.001 | 0.207 +/- 0.001 | 0.071 +/- 0.000 | 0.079 +/- 0.001 | 0.082 +/- 0.001 | 0.036 +/- 0.000 | 0.658 +/- 0.046 |
| | VFE | ✗ | 0.238 +/- 0.000 | 0.341 +/- 0.000 | 0.014 +/- 0.000 | 0.019 +/- 0.000 | 0.030 +/- 0.000 | 0.212 +/- 0.000 | 0.040 +/- 0.000 |
| | FITC | ✗ | 0.302 +/- 0.001 | 0.359 +/- 0.001 | 0.066 +/- 0.001 | 0.068 +/- 0.001 | 0.082 +/- 0.003 | 0.171 +/- 0.002 | 0.000 +/- 0.000 |
| | kl-PAC-SGP | ✓ | 0.115 +/- 0.000 | 0.190 +/- 0.000 | 0.028 +/- 0.000 | 0.034 +/- 0.000 | 0.049 +/- 0.000 | 0.050 +/- 0.000 | 0.254 +/- 0.012 |
| | sqrt-PAC-SGP | ✓ | 0.126 +/- 0.000 | 0.175 +/- 0.000 | 0.054 +/- 0.000 | 0.059 +/- 0.001 | 0.071 +/- 0.000 | 0.027 +/- 0.000 | 0.814 +/- 0.013 |
| | VFE | ✓ | 0.212 +/- 0.000 | 0.327 +/- 0.000 | 0.007 +/- 0.000 | 0.011 +/- 0.000 | 0.024 +/- 0.000 | 0.202 +/- 0.000 | 0.031 +/- 0.000 |
| | FITC | ✓ | 0.277 +/- 0.000 | 0.347 +/- 0.000 | 0.046 +/- 0.000 | 0.048 +/- 0.000 | 0.053 +/- 0.001 | 0.179 +/- 0.000 | 0.000 +/- 0.000 |

Table 3: **Evaluation of inverted Gaussian as loss function** $\ell_{\exp}$. Using the more distance-sensitive loss function $\ell_{\exp}$ from Eq. (20) for our methods "kl-PAC-SGP" and "sqrt-PAC-SGP", we run them against the two standard sparse GP approaches (VFE [6] and VFE [15]) for our three sparse-GP datasets, see Sect. 4. Compare also to Table 2, where the same investigation was done using the 0-1-loss $\ell$ (here, we only report the favorable ARD/non-ARD settings displayed in Fig. 3, cf. Table 2). We again use the following criteria (from left to right): upper bound $B$, Pinsker's upper bound $B_{Pin}$, Gibbs risk on the training data $R_S$[train], Gibbs risk on the test data $R_S$[test], mean squared error (MSE) on the test data, KL-divergence (normalized by the number of training samples, i.e. $KL(Q\|P)/N$), and the learned noise parameter $\sigma_n^2$. The number of inducing inputs is fixed to $M = 500$. We report mean values $\pm$ standard errors over 10 iterations.

| Model configuration | | | Upper bound | | Gibbs risk | | | Model properties | | |
|---|---|---|---|---|---|---|---|---|---|---|
| dataset | method | ARD | $B$ | $B_{Pin}$ | $R_S$[train] | $R_S$[test] | MSE | KL/N | $\sigma^2$ |
| pol | kl-PAC-SGP | ✓ | 0.199 +/- 0.000 | 0.247 +/- 0.000 | 0.077 +/- 0.000 | 0.019 +/- 0.000 | 0.027 +/- 0.000 | 0.041 +/- 0.000 | 0.216 +/- 0.003 |
| | sqrt-PAC-SGP | ✓ | 0.208 +/- 0.001 | 0.245 +/- 0.000 | 0.100 +/- 0.001 | 0.031 +/- 0.001 | 0.040 +/- 0.001 | 0.025 +/- 0.000 | 0.461 +/- 0.006 |
| | VFE | ✓ | 0.288 +/- 0.000 | 0.361 +/- 0.000 | 0.041 +/- 0.000 | 0.006 +/- 0.000 | 0.015 +/- 0.000 | 0.189 +/- 0.000 | 0.016 +/- 0.000 |
| | FITC | ✓ | 0.345 +/- 0.001 | 0.390 +/- 0.001 | 0.085 +/- 0.000 | 0.030 +/- 0.001 | 0.027 +/- 0.000 | 0.169 +/- 0.001 | 0.000 +/- 0.000 |
| sarcos | kl-PAC-SGP | ✗ | 0.116 +/- 0.000 | 0.144 +/- 0.000 | 0.073 +/- 0.000 | 0.012 +/- 0.000 | 0.032 +/- 0.000 | 0.009 +/- 0.000 | 0.213 +/- 0.001 |
| | sqrt-PAC-SGP | ✗ | 0.119 +/- 0.000 | 0.139 +/- 0.000 | 0.086 +/- 0.000 | 0.017 +/- 0.000 | 0.039 +/- 0.000 | 0.005 +/- 0.000 | 0.611 +/- 0.002 |
| | VFE | ✗ | 0.190 +/- 0.000 | 0.259 +/- 0.000 | 0.047 +/- 0.000 | 0.003 +/- 0.000 | 0.017 +/- 0.000 | 0.090 +/- 0.000 | 0.019 +/- 0.000 |
| | FITC | ✗ | 0.227 +/- 0.000 | 0.276 +/- 0.000 | 0.079 +/- 0.000 | 0.015 +/- 0.000 | 0.019 +/- 0.000 | 0.077 +/- 0.000 | 0.000 +/- 0.000 |
| kin40k | kl-PAC-SGP | ✓ | 0.260 +/- 0.001 | 0.290 +/- 0.000 | 0.127 +/- 0.001 | 0.039 +/- 0.001 | 0.055 +/- 0.000 | 0.051 +/- 0.000 | 0.124 +/- 0.014 |
| | sqrt-PAC-SGP | ✓ | 0.262 +/- 0.001 | 0.287 +/- 0.001 | 0.142 +/- 0.001 | 0.048 +/- 0.001 | 0.061 +/- 0.000 | 0.040 +/- 0.000 | 0.232 +/- 0.017 |
| | VFE | ✓ | 0.347 +/- 0.000 | 0.396 +/- 0.000 | 0.076 +/- 0.000 | 0.011 +/- 0.000 | 0.024 +/- 0.000 | 0.202 +/- 0.000 | 0.031 +/- 0.000 |
| | FITC | ✓ | 0.379 +/- 0.000 | 0.413 +/- 0.000 | 0.112 +/- 0.000 | 0.048 +/- 0.000 | 0.053 +/- 0.001 | 0.179 +/- 0.001 | 0.000 +/- 0.000 |

## Footnotes

‡‡One of the 10 iterations ended up in a local optimum with very large $\sigma_n^2$. In contrast to overfitting, this corresponds to underfitting as can be seen by the small value of $KL/N$. Observe also that, within the setting $\varepsilon = 0.2$, the upper bound $B$ is close to 1 for all GPs, indicating a hard prediction problem for the given accuracy of $\varepsilon = 0.2$.