[Reviews · NeurIPS 2018]

Reviewer 1



I have read the replies from reviewers and I decided to increase the score of this paper from 6 to 7. ----- By minimizing PAC-Bayesian Generalization Bounds” the authors propose to train Gaussian processes (GPs) using a PAC-Bayesian bound for bounded loss functions in the regression case. The paper first recalls the classic PAC-Bayesian theorem (see, e.g. McAllester (1999), Theorem 1 and Seeger (2003), Theorem 3.1) for [0, 1]-valued loss func- tions. They show, by using a union bound, that the generalization bound holds for all parameters θ in a finite parameter space Θ and they propose to mini- mize this bound to train GP hyper-parameters in case of full GPs and to select hyper-parameters and inducing points for sparse GP models. In order to use the generalization bound the authors restrict their analysis to the 1-0 loss function. Other loss functions are introduced in appendix. They consider a FITC model (Quiñonero-Candela and Rasmussen, 2005) for sparse GP and train by minimizing the generalization bound. They compare the method with standard methods for full GP and sparse GP training on classic data sets and show that their method gives the best generalization guarantees. Quality: The submission is technically sound. The derivations for the training objectives are well explained and an experimental section details the strengths and weaknesses of the method on standard data sets. Clarity: The submission is clearly written and well organized. Below are some points that might need clarification. • The sparse GP section 3.2 could be improved by clarifying the treatment of the inducing points z 1 , . . . , z M during the training. It is not completely clear in line 195, 196 how and if the discretization of Θ influences inducing points selection. • The experimental section details the methods compared, however gives no indication on the programming language and software used. Nonetheless he authors note that the software implementing the experiments will be made available after acceptance. • line 159 it is not clear to me why a coarser rounding should improve the bound or the optimization. • line 200 the FITC model trained with marginal likelihood is known to over-fit, while the kl-PAC-SGP training objective seems to alleviate this behavior it would be interesting to test this on pathological data sets where FITC is known to fail. Originality: The method proposed, to the best of my knowledge, is new for the task of GP regression. However, the use of [0, 1]-valued loss functions renders it a direct extension of GP classification training. Germain et al. (2016) develop a PAC-Bayesian bound for regression with unbounded loss functions which however requires distributional assumptions on the input examples. The authors mention this and note that in order to keep a distribution-free frame- work they use bounded loss functions. The very interesting question raised by this submission is if it is possible to generalize the training method to the unbounded loss functions commonly used when evaluating regression tasks. Significance: The submission introduces a training method for GP regres- sion which, to the best of my knowledge, is a new technique. By shifting the focus on generalization guarantees it provides an interesting point of view in the likelihood dominated landscape of GP training. However, in my opinion, there are several aspects that reduce the significance of this work. • the use of [0, 1]-valued loss functions makes this training method much less appealing for practitioners as it leads to less accurate results in MSE than classic methods. • Germain et al. (2016) show that if we consider as loss function a bounded negative log-likelihood then minimizing the PAC-Bayesian bound is equiv- alent to maximizing the marginal likelihood. I am wondering if by using an appropriately bounded negative log-likelihood we could obtain better results than [0, 1]-valued loss functions in terms of upper bound and Gibbs risk. • The authors show that the discretization of the parameter space Θ does not affect the method drastically on an example. In the same section, Ap- pendix F, they mention that “as long as a minimal discrimination ability is allowed” (line 502) the training and test risks are not affected. However I am not fully convinced that for ARD kernels with high d a minimal dis- crimination ability is within reach computationally because the number of parameter combinations increases drastically.

Reviewer 2



This paper proposes to directly optimize the PAC-Bayesian bound of Gaussian process regression using a 0/1 loss with respect to the hyperparameters and the inducing points (for sparse GP). The tight version of the bound using the binary KL-divergence is used. The hyper-parameter space must be a-priori discretised for their approach to work. Experiments compare their approach with full-GP and VFE. [Quality] In general, I find this paper to be of sound quality. However, I have some concerns: 1. In section 3.1, the regression form of GP is used --- this is using the Gaussian noise model, which directly targets the MSE of the prediction. However, the posterior under this noise model is then used as the $Q$ with the 0/1 loss. There seems to be some mismatch here --- the first level inference uses MSE, while the second level inference uses the 0/1 loss. For example, I should think that a more appropriate Bayesian inference model would be to use variational inference with a uniform noise model. 2. In section 3.2, the authors suggest to optimize over $\alpha$. However, this can potentially cause variances to be underestimated during prediction, since FITC is known to sometimes give negative variance. This aspect has not been addressed. For example, in Figure 1, is the sometimes smaller variance of kl-PAC-SGP caused by this? What are the $\alpha$s learnt? What if $\alpha$ is fixed at 0 instead of learnt? Not enough analysis is done. 3. Since this is a PAC-Bayesian analysis, comments on the experimental results should also address the tightness of the bounds. 4. There should be comments and analysis on why both kl-PAC-GP/SGP is worse than Full-GP/VFE in terms of both Gibbs test risk and MSE. 5. Eq 32 should use some probabilistic bounding between $R_S(Q)$ and the mini-batch average and then reinsert these into the PAC-Bayesian bound. In any case, if we use exactly $B$ data points, shouldn't the bound be on $B$ data points and not $N$? More justification other than efficiency is needed here. [Clarity] L154. Please define $T$ before its use. What is the objective used for the gradient minimization? Since the objective cannot depend on the training data, it seems to be neither ML-II nor PAC-Bayesian bound? Also, please elaborate on "minimization to the closest point in the equispaced...". Footnote 4, last sentence. In what way do we "need" the noise parameter? Do you mean we need to optimize for it and bound it away from zero? Experiment 4a. It is not totally clear what are learnt and what are fixed. I assume that the hyperparameters are fixed in this exercise. Experiment 4b. I find no need to compare with sqrt-PAC-GP in the main paper --- suggest to move this to the supplementary. [Originality] This paper provides another objective for hyper-parameter optimization for GP learning, [Significance] 1. The paper is primarily for bounded loss, so it is not directly application to MAE and MSE which are more natural and commonly used for regression. 2. The method does not do better than VFE for MSE, albeit it does not optimize directly for MSE during hyper-parameter optimization. In addition, it does not also do better for the Gibbs test risk. [Minor] L3: Do you mean "tight" for "good"? L41: Add [10] as a reference for the binary KL divergence. Footnote 4, last sentence: "diverges" and 'vanishes" are redundant t. In what way do we "need" the noise parameter? Do you mean we need to optimize for it and bound it away from zero? L196-L198: The sentence does not seem to be well-formed. L200: PAC-SPG [Comments after authors reply] I am satisfied with the reply and look forward to the authors addressing some of the points noted above in the paper.

Reviewer 3



Summary of the Paper: This paper describes a new method of training Gaussian process for regression based on minimizing a bound on the generalization error. This approach can have advantages over others in the sense that it will provide generalization guarantees, which may be of practical importance in some applications. The proposed approach is evaluated in several regression tasks and compared with popular approaches for GP regression based on FITC or VAE. Strengths: - Well written paper. - Complete related work section. - Nice and illustrative experiments. Weaknesses: - The MSE of the proposed approach is worse than the one obtained by FITC of VFE. Quality: I think the quality of the paper is high. It addresses an important problem and the methodological steps and evaluation procedure are right. Originality: As far as I know this paper is original. I am not aware of other works trying to fit GPs by minimizing a generalization bound. Significance: The experiments confirm the utility of the proposed approach. A disadvantage is, however, that it performs worse in terms of MSE than VFE or FITC. Notwithstanding, it is normal since a different objective is being optimized here. Other comments: Summing up, I believe that this is an interesting paper that, although does not provide outstanding results, may attract the attention of the ML community working on GPs. After reading the authors response and viewing the comments of the other reviewers, I have slightly increased my score.